# No evidence of positive causal effects of maternal and paternal age at first birth on children's test scores at age 10 years

**Michael Grätz** [1,2,8] ✉, **Felix C. Tropf** [3,4,8] ✉, **Fartein Ask Torvik** [5], **Ole A. Andreassen** [6] **& Torkild H. Lyngstad** [7,8] ✉

Research has shown that higher maternal and paternal age is positively associated with children's education. Debate continues as to whether these relationships are causal. This is of great interest given the postponement of first births in almost all developed countries during the twentieth century. Here we use an instrumental variable approach (Mendelian randomization) using maternal and paternal polygenic indices (PGIs) for age at first birth—while conditioning on the child's PGI for age at first birth—to identify the causal effects of maternal and paternal age at first birth on children's test scores based on data from the Norwegian Mother, Father and Child Cohort study. We do not find evidence of positive causal effects of both maternal and paternal age at first birth on children's test scores at age 10 years once the children's PGI and correlations among different PGIs are controlled for. We therefore conclude that our results do not provide evidence in favour of sociological theories that predict positive causal effects of parental age on children's educational attainment.

During the second half of the twentieth century, women's age at first birth (AFB) increased by about 4–5 years in almost all developed countries[1]. The consequences of this demographic shift are of great interest to policymakers, the public and academic researchers across a range of scientific disciplines. At the individual level, parents and their offspring may face a trade-off between the negative biological and the positive socioeconomic consequences of postponing parenthood. At the population level, these processes may be of great importance for disease prevalence and social stratification.

Children of older parents are at higher risk of being born with a low birth weight and of developing symptoms of mental-health conditions such as autism and schizophrenia[2]. However, there are positive associations between older parental age and various measures of children's

educational success[3–6]. In the United States, for example, each one-year increase in maternal age is associated with a 0.039 standard deviation (s.d.) increase in children's mathematics scores[4].

The key question is whether the associations between maternal and paternal age and children's educational outcomes are causal. Only if maternal and paternal age have a causal effect on children's educational outcomes will a shift in the distribution of paternal or maternal age lead to improved educational outcomes at the population level.

Previous research has mainly relied on family fixed-effects models to identify the causal effects of maternal and paternal age on children's educational outcomes[3–5]. These studies found similar effect sizes within and across families. The interpretation of these estimates as causal has been questioned because of the possibility of unobserved confounding

[1]Swiss Centre of Expertise in Life Course Research (LIVES), University of Lausanne, Lausanne, Switzerland. [2]Swedish Institute for Social Research (SOFI), Stockholm University, Stockholm, Sweden. [3]Centre for Longitudinal Studies, Social Research Institute, University College London, London, UK. [4]Department of Sociology, Purdue University, West Lafayette, IN, USA. [5]Centre for Fertility and Health, Norwegian Institute of Public Health, Oslo, Norway. [6]NORMENT Center, Faculty of Medicine, University of Oslo, Oslo, Norway. [7]Department of Sociology and Human Geography, University of Oslo, Oslo, Norway. [8]These authors contributed equally: Michael Grätz, Felix C. Tropf, Torkild H. Lyngstad. ✉e-mail: michael.gratz@unil.ch; ftropf@purdue.edu; t.h.lyngstad@sosgeo.uio.no

variables that vary between siblings. In addition, it is unclear whether one should adjust for the offspring's year of birth. Adjusting for the birth year of the offspring is problematic because it is perfectly collinear with parental age in a family fixed-effects model. However, not adjusting for birth year to avoid the multicollinearity problem results in any parental age effect potentially being driven by the difference in the time period in which different siblings are born[7,8].

Here we approach the question of the causal effects of parental age on children's education using a genetic instrumental variable (IV) design based on Mendelian randomization (MR)[9]. Specifically, our approach enables us to estimate the causal effects of maternal and paternal AFB on children's test scores at age 10 years. We use maternal and paternal PGIs for AFB as instruments and condition on the child's PGI for AFB.

We identify three challenges to the MR approach from recent genetic literature, which we address using the Norwegian Mother, Father and Child Cohort study (MoBa), a Norwegian pregnancy cohort linked to registry data and genotype data on trios. First, the same genetic variants may predict multiple outcomes. For example, AFB-linked genetic variants co-predict the risk of externalizing behaviour and educational attainment[10]. This phenomenon, known as pleiotropy, can be due to shared biological influences, social mediation or genetic effects on third common causes[11]. This effect can be more pronounced the further a predicted outcome is from actual biological processes, which makes AFB particularly susceptible to pleiotropy. Pleiotropy leads to a violation of the exclusion restriction assumption that is central to the IV approach, according to which the instrument does not influence the outcome other than through the instrumented variable. We therefore control for PGIs of heritable AFB predictors in the parental generation. Specifically, we control for sexual, contraceptive and smoking behaviour, attention-deficit/hyperactivity disorder (ADHD) and educational attainment[10].

Second, because we are interested in the effect of paternal and maternal AFB on children's test scores, and children inherit 50% of each parent's DNA, genetic inheritance could bias the results if genes that affect parental AFB directly influence children's test scores. Therefore, we jointly include parental and child PGIs for AFB in the statistical model. In this way, we minimize the possibility that observed associations between parental and child outcomes are due to shared, instrument-related genotypes.

Third, genetic discoveries consist of a series of simple regression models with few demographic and genetic control variables, and may be confounded by environmental correlates, assortative mating (AM) or population stratification[12]. These confounding effects could lead to an inflation of causal estimates relative to standard regressions. We control for the PGI of the partner's AFB to account for parental AM[13]. To account for grandparental AM leading to potential excess homozygosity, we removed single nucleotide polymorphisms (SNPs) that are out of the Hardy–Weinberg equilibrium based on a $P$ value threshold of $1 \times 10^{-6}$ (ref. 14). In addition, we control for the respective PGI of inherited SNPs to account for potentially remaining excess homozygosity. To maximize our control for genetic nurture effects and population stratification, we consider the first ten principal components and the education level of the offspring's grandparents. Finally, use of Norwegian registry data enabled us to avoid several recently highlighted issues of sample selection and resulting ascertainment bias[15].

We graphically represent our research design in a directed acyclic graph in Fig. 1.

## Results

The results of the IV models are reported in Table 1. We report estimates, obtained in separate models, for both maternal and paternal AFB. We also report results for the corresponding cross-sectional ordinary least squares (OLS) regression models.

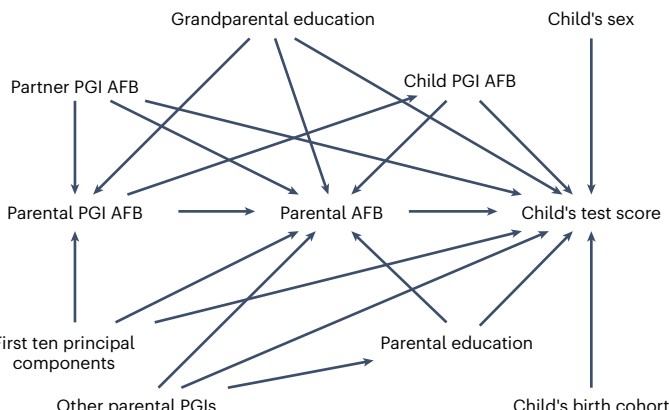

**Fig. 1 | Our IV approach portrayed in the form of a directed acyclic graph.** Parental PGI AFB is the instrumental variable for parental AFB. The outcome is the child's test score. To satisfy the independence and exclusion restriction assumption, we introduced several control variables. The following parental PGIs were included: (1) parental PGI age at first sexual intercourse; (2) parental PGI for smoking (3) parental PGI for contraception use; (4) parental PGI for ADHD; and (5) parental PGI for education.

The OLS models showed positive associations between both maternal and paternal AFB and children's education, a result consistent with previous research[3–6]. Without control variables, a 1-year delay in the mother's AFB was associated with 3.7% of a s.d. (confidence interval (CI): 3.4%, 4.0%) increase in test scores in fifth grade (around age 10) (model 1 OLS). This association was slightly smaller for the father's AFB, with 2.3% of a s.d. (CI: 2.0%, 2.5%) increase in achievement (model 1 OLS). Adding the control variables and parental PGI for education reduced the estimates to 1.4% (CI: 0.9%, 1.8%) for mothers and 0.9% (CI: 0.6%, 1.2%) for fathers (models 13 OLS). These associations cannot be interpreted causally because of the potential remaining unobserved confounding.

The first-stage $F$ statistics of the IV models were substantial, and our analyses did not suffer from weak instrument bias. The IV models without our control variables showed a strong inflation of the estimates compared with the OLS models for both maternal (12.6%; CI: 10.5%, 14.7%) and paternal (19.8%; CI: 15.6%, 23.9%) AFB (model 1 IV). However, once we included the control variables in the IV models, the effects became nonsignificantly different from 0 (model 12 IV). Once we added a control for the parental educational attainment PGI, the estimates turned around, and we even found evidence for a negative causal effect of AFB on test scores at age 10 in the full model. In detail, the estimates for both maternal and paternal AFB become significantly negative (mothers: −8.8% (CI: −15.5%, −2.1%); fathers: −19.3% (CI: −35.1%, −3.6%); model 13 IV). Although the inclusion of the PGI of education may have reduced bias by controlling for a confounder, it could also have introduced overcontrol bias if the PGI for education picked up part of the effect that actually belongs to the PGI for AFB. The correlation between the PGI for AFB and the PGI for education is shown in Supplementary Table 3. We therefore conservatively conclude that our results do not provide support for the idea that parental age positively affects children's educational performance.

To unpack these results, we analysed our model specifications in detail, performing analyses with covariates added stepwise (Fig. 2). We observed two major shifts in the IV estimates across specifications. First, the full PGI of AFB for both parents were positively predictive of children's test scores when both indirect genetic effects and inherited alleles were combined. Once the model was controlled for the child's PGI of AFB and identified nurture and ancestry effects for the instrument, respectively, the estimate approached zero. Second, inclusion of the parental PGI for educational attainment turned the estimates negative. In summary, our conservative conclusion is that there is no

**Table 1 | Estimates of the effects of maternal and paternal AFB on children's test scores at age 10**

| Model type | OLS | | | | IV | | | |
|---|---|---|---|---|---|---|---|---|
| Model number | 1 | 2 | 12 | 13 | 1 | 2 | 12 | 13 |
| Maternal (*n*=15,670) | | | | | | | | |
| Estimate | 0.037 | 0.037 | 0.015 | 0.014 | 0.126 | 0.124 | −0.002 | −0.088 |
| 95% CI | 0.034, 0.040 | 0.034, 0.040 | 0.012, 0.019 | 0.009, 0.018 | 0.105, 0.147 | 0.103, 0.145 | −0.051, 0.046 | −0.155, −0.021 |
| *F* statistic | | | | | 417 | 432 | 74.1 | 47.9 |
| BIC | 39,960 | 40,033 | 37,307 | 37,199 | 43,003 | 42,946 | 37,416 | 40,459 |
| Paternal (*n*=15,593) | | | | | | | | |
| Estimate | 0.023 | 0.0221 | 0.009 | 0.009 | 0.198 | 0.193 | −0.011 | −0.193 |
| 95% CI | 0.020, 0.025 | 0.019, 0.025 | 0.006, 0.012 | 0.006, 0.012 | 0.156, 0.239 | 0.152, 0.233 | −0.107, 0.085 | −0.351, −0.036 |
| *F* statistic | | | | | 144 | 146 | 13.7 | 11.3 |
| BIC | 40,258 | 40,287 | 36,976 | 36,777 | 51,135 | 50,724 | 37,161 | 49,580 |
| Model specification | | | | | | | | |
| PCs 1–10 | No | Yes | Yes | Yes | No | Yes | Yes | Yes |
| Controls | No | No | Yes | Yes | No | No | Yes | Yes |
| Parental PGI for education | No | No | No | Yes | No | No | No | Yes |

The outcome is measured in s.d. values and the exposure in years. The table reports the point estimates and the 95% confidence intervals. Controls include grandparental education, partner's PGI for AFB, parental education, child PGI for AFB, child's birth cohort (continuous), child's sex, parental PGI age at first sexual intercourse, parental PGI for smoking, parental PGI for contraception use, and parental PGI for ADHD. The *F* statistic is the partial effect of the instrument on the exposure. BIC in the IV models refers to the 2SLS IV model. See Fig. 2 for model specifications.

robust evidence of positive effects of paternal and maternal AFB on children's test scores at age 10 in Norway.

## Discussion

Numerous studies have reported positive associations between maternal and paternal age and children's educational outcomes, even within families[3–6]. However, it has been unclear whether these positive associations are due to underlying positive causal effects of higher maternal and paternal age. Using an MR approach, we did not find robust evidence for positive causal effects of maternal and paternal AFB on children's test scores at age 10.

Specification analyses revealed that the positive association was due to a null effect once the children's PGI and correlations between different PGIs were taken into account. In an IV model without further controls, we observed a large positive effect of parental age, which we consider implausible and unreliable. This effect is probably due to unobserved confounding. Controlling for children's PGI for AFB reduced the IV estimate. Finally, we observed a strong change in the IV estimate when we included the PGI for parental educational attainment. This is unsurprising as the genetic association between the PGI for education and the PGI for AFB is well established, and the education PGI is known to be one of the strongest PGIs in behavioural science. However, at the same time, the PGI for education is a proxy measure, which captures all factors related to education, including diseases, cognitive skills and personality traits. For this reason, conditioning on the PGI for education could also introduce overcontrol bias.

Although IV approaches in general, and MR approaches in particular, have been challenged for potential violations of assumptions such as the independence assumption, exclusion restriction or population stratification[12], we think it is unlikely that the effects of such violations perfectly balance our results towards zero or are responsible for making them negative in the full model. However, we cannot rule out potential biases, particularly due to pleiotropy. Recruitment bias may have been introduced in our study because the data from the genome-wide association study (GWAS) used in our study were largely from the UK Biobank, which had a low participation rate and is known to be an imperfect reflection of the general population. However, it is unclear whether this biases our estimates with respect to the specific research question we analysed.

Our results do not support sociological expectations of positive effects of higher maternal and paternal age on children's education. The positive associations between parental ages and children's academic performance have been attributed by social scientists to higher socioeconomic status and the greater resources that parents accumulate as they age[6]. In addition, it has been argued that higher parental age leads to more stable relationships and better parenting[5], which is beneficial for children's academic performance. Our results do not provide evidence to support these expectations, although uncertainty remains owing to the large CI values.

Our findings also do not support biological theories that predict even negative effects of paternal and maternal age. Biologists have observed potentially deleterious mental-health consequences of increasing parental ages, particularly related to the paternal accumulation and inheritance of de novo mutations across the life course and increased psychiatric problems in children. In addition, age-related methylation changes[16], which may modify parenting behaviour, may influence children's school performance, including episodes of psychosis[17–19]. Although our results do not directly address these potential mechanisms influencing children's test scores, and mathematical models have shown that de novo mutations are unlikely to have large effects in general[20], they do not provide evidence for robust negative effects of parental age. The possibility remains that positive social consequences of parental age effects and negative biological consequences cancel each other out.

The MR approach has been promoted[21] and challenged[22] in the social sciences. Our study showed that its naive use can lead to misleading, inflated and implausible results. The quantitative solutions to challenges in genetic discovery and the genetic architecture of human behaviour—in particular, polygenicity and pleiotropy—suggest that applications can be informative as complements to the literature.

Our results were obtained with respect to a specific society. Norway has a high per capita income compared with other developed countries, and has a low level of income inequality. In addition, the Norwegian government provides generous child benefits. For this

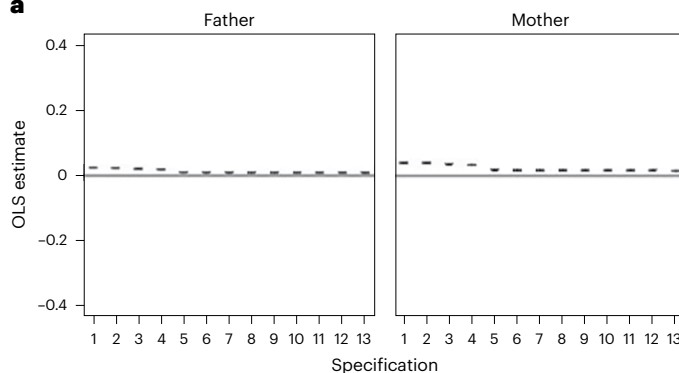

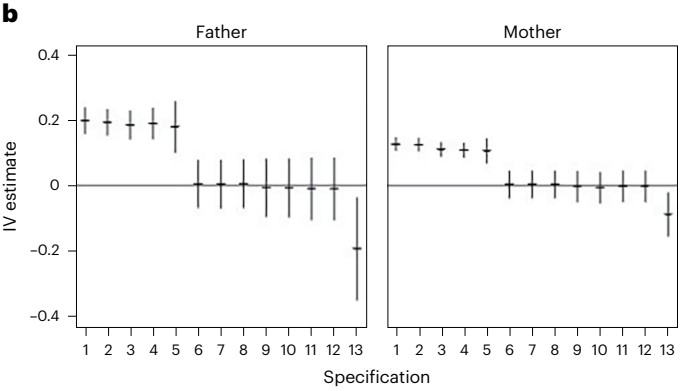

**Fig. 2 | The effects of maternal and paternal ages AFB on children's test scores at age 10 (s.d. of educational attainment per year of AFB), estimates from different model specifications with covariates added stepwise. a,b,** Estimates obtained for OLS (**a**) and IV (**b**) models for both maternal and paternal AFB. The following control variables were included stepwise: (1) none; (2) first ten genetic principal components; (3) grandparental education; (4) partner's PGI for AFB; (5) parental education; (6) child PGI for AFB; (7) child's birth cohort (continuous); (8) child's sex; (9) parental PGI age at first sexual intercourse; (10) parental PGI for smoking; (11) parental PGI for contraception use; (12) parental PGI for ADHD; and (13) parental PGI for education. Plots show point estimates and CIs; the CIs are very small for the OLS estimates. *n* = 15,670 (maternal) and 15,593 (paternal) for both **a** and **b**.

reason, the positive causal effect of parental age on child education may be smaller in Norway than in other societies, such as the United States. Our results are also specific to the analysis of the AFB, and other research designs are needed to study the effects of parental age at higher order births.

It is important to highlight both the advances our data provide for our modelling approach compared with standard MR approaches, as well as the remaining limitations. In particular, the MoBa data enabled us to account for AM. However, the correction for AM, which controls for partner PGI, remains incomplete owing to missing heritability, which may potentially inflate causal estimates. We also controlled for population stratification, including genetic principal components and grandparental education. However, these controls also remain incomplete, and future studies should aim to use results from within-family GWAS or perform within-family prediction[23] when such data are available.

Finally, given the uncertainty in our estimates and the fact that we provided only one specific modelling approach to causality, we do not provide a definitive answer but add new evidence to the literature that contradicts the idea that parental age leads to better education for children, at least independent of parental education and other correlated factors that should be taken into account. Our results also implied that the effects of postponing fertility at the population level largely depend on age-related conditions that seem to compensate

for potential negative effects of ageing, all of which can be replicated in even larger studies using different research designs. It is still possible that social and biological processes act in opposite directions, as predicted by the theories, but cancel each other out. At the macro level, this interpretation implies that increasing AFB will not lead to improved educational outcomes at the population level.

## Methods

### Data

We used data from MoBa[14,24], which is a population-based pregnancy cohort study conducted by the Norwegian Institute of Public Health. Participants were recruited from all over Norway from 1999 to 2008. The women consented to participate in 41% of cases. No compensation was paid. The cohort includes approximately 114,500 children, 95,200 mothers and 75,200 fathers. The current study is based on version 12 of the quality-assured data files, which was released for research in 2019. The establishment of MoBa and the initial data collection were based on a licence from the Norwegian Data Protection Agency and approval from the Regional Committees for Medical and Health Research Ethics. Data from the MoBa cohort are regulated by the Norwegian Health Registry Act. Our sample was restricted to first-born children born in Norway between 2001 and 2008 to focus on the causal effects of maternal and paternal ages AFB. The sample size is 15,670 for the models using maternal AFB and 15,593 for the models using paternal AFB. Given these sample sizes and an alpha set to the standard value of 0.05, the power of our analysis to identify an effect size of 0.034 s.d. (OLS estimate in model 1) is 0.99 for the IV analysis. Overall, 50.7% of the children in the sample are female.

We report the first-stage results in Supplementary Table 1 and the reduced-form estimates in Supplementary Table 2. Supplementary Table 3 reports a correlation matrix among the variables included in our analysis.

### Phenotypic measures

We measured children's academic performance using national standardized tests in mathematics, reading and English in fifth grade (around age 10). We analysed a measure that averages across subjects. If there were missing values in one subject, we averaged using the other test scores. We standardized test scores within each individual and birth year to have a mean of 0 and a s.d. of 1. For the AFB, we included only children for whom both parents registered as first-time parents.

### Genetic measures

Genotyping of MoBa has been performed in different projects with various selection criteria, genotyping arrays and genotyping centres[13]. Phasing and imputation have been performed using the publicly available Haplotype Reference Consortium release 1.1 panel as a reference. The MoBaPsychGen pipeline, comprising pre-imputation quality control (QC) based on SNPs and individuals, phasing, imputation and post-imputation QC based on current best-practice protocols, was implemented to account for the complex (relatedness) structure of the data. In total, 6,981,748 SNPs passed through the pipeline. Our PGI constructions were based on all these SNPs. Each PGI construction was based on summary statistics provided by the trait-specific GWAS.

### Analytical strategy

We used a MR approach to estimate the causal effects of parental age on a child's education[9]. We instrumented maternal and paternal AFB using genes associated with AFB in a previous genetic discovery study[10] and summed the results to create a PGI to ensure sufficient statistical power. We use the parental PGI for AFB as the instrument while conditioning on the child's PGI for AFB. We also considered population stratification controlling for grandparental education and the first ten genetic principal components, AM controlling for partner's PGI for AFB, demographic

characteristics (birth year and child's sex), and controlling for the PGIs for age at first sexual intercourse, age at smoking initiation, age at first use of oral contraceptives, ADHD, and educational attainment, which have previously been linked to the PGI for AFB[10] and may influence parental nurturing behaviour relevant to child test scores. All statistical tests in our regression models are two-sided. For comparison reasons, we also report OLS regression models that estimate the associations between parental age and children's test scores, without controlling for the endogeneity of AFB.

## Reporting summary

Further information on research design is available in the Nature Portfolio Reporting Summary linked to this article.

## Data availability

The MoBa data are available under restricted access owing to data-protection laws, and access can be obtained by application to MoBa and a Regional Committee for Medical and Health Research Ethics in Norway. If you wish to apply for access, please visit the following website for more details: https://www.fhi.no/en/ch/studies/moba/for-forskere-artikler/research-and-data-access/.

## Code availability

All analyses were conducted using R. All code to replicate the analyses is available from GitHub (https://github.com/torkildl/parental-age).

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

## Acknowledgements

M.G. was supported by the Swiss National Science Foundation (SNSF) under grant agreement TMSGI1_211627 and by the Forskningsrådet om Hälsa, Arbetsliv och Välfärd (Forte) (2016–07099). F.C.T. was supported by the UKRI project FINDME (EP/Y023080/1) and the Laboratory of Excellence in Economics and Decision Sciences (LabEx Ecodec) funded under the French National Research Agency (ANR) Investissements d'Avenir (ANR-11-LABX-0047). F.A.T. was supported by the Research Council of Norway (334093). This work was partly supported by the Research Council of Norway through its Centres of Excellence funding scheme (262700). T.H.L. was supported by funding from the European Research Council (ERC) under the European Union's Horizon 2020 research and innovation programme under grant agreement number 818420 (OPENFLUX). MoBa is supported by the Norwegian Ministry of Health and Care Services and the Ministry of Education and Research. We are grateful to all the participating families in Norway who have contributed to this ongoing cohort study. We thank the Norwegian Institute of Public Health (NIPH) for generating high-quality genomic data. This research is part of the HARVEST collaboration, supported by the Research Council of Norway (229624). We also thank the NORMENT Center for providing genotype data, funded by the Research Council of Norway (223273), South-Eastern Norway Regional Health Authority and Stiftelsen Kristian Gerhard Jebsen. We further thank staff at the Center for Diabetes Research, the University of Bergen for providing genotype data and performing quality control and imputation of the data funded by the ERC AdG project SELECTionPREDISPOSED, Stiftelsen Kristian Gerhard Jebsen, Trond Mohn Foundation, the Research Council of Norway, the Novo Nordisk Foundation, the University of Bergen, and the Western Norway Health Authorities. We are grateful to A. Havdahl, E. Corfield and staff at the MobaPsychGen project for access to quality controlled genotype data. The funders had no role in study design,

data collection and analysis, decision to publish or preparation of the manuscript. The Norwegian registry and MoBa data used was from the project SUBPU. The Department of Psychology, University of Oslo, is responsible for the data handling of SUBPU, a Data Protection Impact Assessment (DPIA) has been signed by the head of department, and the project manager is Eivind Ystrøm. SUBPU is approved by Committees for Medical and Health Research Ethics (#2017/2205). SUBPU has agreements with the MoBa and the Statistics Norway for data linkage and usage. The data access and management costs of SUBPU is financed by the Research Council of Norway (RCN) (#336078, #288083, and #314601), the European Research Council (#101045526, #818425, #101088481, and #818420), and supported by the Department of Psychology (UiO). All data management and analyses were on the secure data "Tjeneste for Sensitive Data" (TSD) facilities, owned by the University of Oslo. Resources provided by Sigma2, the National Infrastructure for High Performance Computing and Data Storage in Norway (UNINETT), was used for analyses (#NS9867S).

## Author contributions

M.G., F.C.T. and T.H.L. conceptualized the study and research design. T.H.L. conducted the analyses. M.G. and F.C.T. wrote the paper. F.A.T. and O.A.A. provided substantial feedback and revisions on the article. M.G. and F.C.T. coordinated revisions and submission. All authors reviewed and approved this article before submission. All authors read and agreed to the published version of the article.

## Funding

## Competing interests

The authors declare no competing interests.

## Additional information

**Correspondence and requests for materials** should be addressed to Michael Grätz, Felix C. Tropf or Torkild H. Lyngstad.

| | Corresponding author(s): | Michael Grätz |
|---|---|---|
| | Last updated by author(s): | 2024/11/17 |

# Reporting Summary

## Statistics

For all statistical analyses, confirm that the following items are present in the figure legend, table legend, main text, or Methods section.

| n/a | Confirmed | |
|---|---|---|
| ☐ | ☒ | The exact sample size ($n$) for each experimental group/condition, given as a discrete number and unit of measurement |
| ☐ | ☒ | A statement on whether measurements were taken from distinct samples or whether the same sample was measured repeatedly |
| ☐ | ☒ | The statistical test(s) used AND whether they are one- or two-sided<br>*Only common tests should be described solely by name; describe more complex techniques in the Methods section.* |
| ☐ | ☒ | A description of all covariates tested |
| ☐ | ☒ | A description of any assumptions or corrections, such as tests of normality and adjustment for multiple comparisons |
| ☐ | ☒ | A full description of the statistical parameters including central tendency (e.g. means) or other basic estimates (e.g. regression coefficient) AND variation (e.g. standard deviation) or associated estimates of uncertainty (e.g. confidence intervals) |
| ☐ | ☒ | For null hypothesis testing, the test statistic (e.g. $F$, $t$, $r$) with confidence intervals, effect sizes, degrees of freedom and $P$ value noted<br>*Give P values as exact values whenever suitable.* |
| ☒ | ☐ | For Bayesian analysis, information on the choice of priors and Markov chain Monte Carlo settings |
| ☒ | ☐ | For hierarchical and complex designs, identification of the appropriate level for tests and full reporting of outcomes |
| ☒ | ☐ | Estimates of effect sizes (e.g. Cohen's $d$, Pearson's $r$), indicating how they were calculated |

*Our web collection on statistics for biologists contains articles on many of the points above.*

## Software and code

Policy information about availability of computer code

| Data collection | NA |
|---|---|
| Data analysis | Data analysis were conducted using R. The R code to replicate all analyses is available at https://github.com/torkildl/parental-age. |

For manuscripts utilizing custom algorithms or software that are central to the research but not yet described in published literature, software must be made available to editors and reviewers. We strongly encourage code deposition in a community repository (e.g. GitHub). See the Nature Portfolio guidelines for submitting code & software for further information.

## Data

Policy information about availability of data

All manuscripts must include a data availability statement. This statement should provide the following information, where applicable:
- Accession codes, unique identifiers, or web links for publicly available datasets
- A description of any restrictions on data availability
- For clinical datasets or third party data, please ensure that the statement adheres to our policy

The MoBa data is available under restricted access due to data protection laws, and access can be obtained by application to MoBa and a Regional Committee for Medical and Health Research Ethics in Norway. If you wish to apply for access, please visit the following website for more details: https://www.fhi.no/en/ch/studies/moba/for-forskere-artikler/research-and-data-access/.

# Research involving human participants, their data, or biological material

Policy information about studies with human participants or human data. See also policy information about sex, gender (identity/presentation), and sexual orientation and race, ethnicity and racism.

| | |
|---|---|
| Reporting on sex and gender | Mothers and Fathers are analysed seperately in our study. Sex has been self-reported. |
| Reporting on race, ethnicity, or other socially relevant groupings | NA |
| Population characteristics | Our regression models control of age of the participants, grandparental education, population stratification using genetic principal components, polygenic scores for age at first birth, age at first sexual intercourse, age at smoking initiation, age at first use of oral contraceptives, attention deficit hyperactivity disorder, educational attainment, for age and sex of the children. |
| Recruitment | The data consists of a population based cohort study conducted by the Norwegian Institute of Public Health. Pregnant women were recruited in Norway from 1999 to 2008. Women consented to participate in 41% of cases. All participants consented. No compensation was paid. |
| Ethics oversight | Regional Committee for Medical and Health Research Ethics in Norway |

Note that full information on the approval of the study protocol must also be provided in the manuscript.

# Field-specific reporting

Please select the one below that is the best fit for your research. If you are not sure, read the appropriate sections before making your selection.

☐ Life sciences   ☒ Behavioural & social sciences   ☐ Ecological, evolutionary & environmental sciences

For a reference copy of the document with all sections, see nature.com/documents/nr-reporting-summary-flat.pdf

# Behavioural & social sciences study design

All studies must disclose on these points even when the disclosure is negative.

| | |
|---|---|
| Study description | We conduct a quantitative Mendelian Randomization analyses using IV regression models. |
| Research sample | We use existing data from the Norwegian Mother, Father, Child Cohort Study, a population based sample of pregnant women in Norway conducted from 1999 to 2008. This study has both parents and the children resulting from the pregnancy genotyped which allowed us for some statistical modeling that is quite unique to our study. |
| Sampling strategy | All pregnant women attending a routine ultrasound examination were invited by the data collectors. We did not collect data ourselves. |
| Data collection | Blood samples were collected from fathers and mothers and questionnaires were sent to both parents. Records were linked to existing registry data. |
| Timing | 10/99 - 07/08 |
| Data exclusions | Of all MoBa children (114500), we used information on those children (17523) who themselves were genotyped, had genotyped parents and were the first-born child of both parents. |
| Non-participation | The response rate was 41% |
| Randomization | There was no randomization, it's a population-based data set. We used multivariate statistical models for causal inference. |

# Reporting for specific materials, systems and methods

We require information from authors about some types of materials, experimental systems and methods used in many studies. Here, indicate whether each material, system or method listed is relevant to your study. If you are not sure if a list item applies to your research, read the appropriate section before selecting a response.

## Materials & experimental systems

| n/a | Involved in the study |
|-----|----------------------|
| ☒ | Antibodies |
| ☒ | Eukaryotic cell lines |
| ☒ | Palaeontology and archaeology |
| ☒ | Animals and other organisms |
| ☒ | Clinical data |
| ☒ | Dual use research of concern |
| ☒ | Plants |

## Methods

| n/a | Involved in the study |
|-----|----------------------|
| ☒ | ChIP-seq |
| ☒ | Flow cytometry |
| ☒ | MRI-based neuroimaging |

