## [Peer Review File · Nature Human Behaviour]

No Evidence of Positive Causal Effects of Maternal and Paternal Age at First Birth on Children's Test Scores at Age 10

Corresponding Author: Professor Michael Grätz

Version 0:

Decision Letter:

28th November 2023

Dear Dr Tropf,

Thank you once again for your manuscript, entitled "No Evidence for Causal Effects of Maternal and Paternal Ages at First Birth on Children's Test Scores at Age 10", and for your patience during the peer review process.

Your Article has now been evaluated by 3 referees. You will see from their comments copied below that, although they find your work of considerable potential interest, they have raised quite substantial concerns. In light of these comments, we cannot accept the manuscript for publication, but would be interested in considering a revised version if you are willing and able to fully address reviewer and editorial concerns.

We hope you will find the referees' comments useful as you decide how to proceed. If you wish to submit a substantially revised manuscript, please bear in mind that we will be reluctant to approach the referees again in the absence of major revisions. We are committed to providing a fair and constructive peer-review process. Do not hesitate to contact us if there are specific requests from the reviewers that you believe are technically impossible or unlikely to yield a meaningful outcome.

1) The reviewers - and especially Reviewer #1 - raise several fundamental concerns about your analytical approach and recommend numerous additional or alternative analyses that we ask you perform in full.

2) The reviewers also ask for several clarifications and additional information on your methods in order to enable robust evaluation of your work. Please ensure that all this additional information is provided.

3) Editorially, for manuscripts where the main result reported is null, we ask that you:

- a) Provide Bayes Factors that quantify support for the null hypothesis.
- b) Carry out power sensitivity analyses and demonstrate that your sample is sufficiently powered to capture the smallest effect size of theoretical/practical interest.

Please keep in mind that null results in underpowered samples are not meaningful or interpretable.

If you wish to submit a suitably revised manuscript, we would hope to receive it within 3 months. I would be grateful if you could contact us as soon as possible if you foresee difficulties with meeting this target resubmission date.

- Include a "Response to the editors and reviewers" document detailing, point-by-point, how you addressed each editor and referee comment. If no action was taken to address a point, you must provide a compelling argument. When formatting this document, please respond to each reviewer comment individually, including the full text of the reviewer comment verbatim followed by your response to the individual point. This response will be used by the editors to evaluate your revision and sent back to the reviewers along with the revised manuscript.
- Highlight all changes made to your manuscript or provide us with a version that tracks changes.

Link Redacted

Thank you for the opportunity to review your work. Please do not hesitate to contact me if you have any questions or would like to discuss the required revisions further.

Sincerely,

Giacomo Ariani
Editor
Nature Human Behaviour

Reviewer expertise:

Reviewer #1: Mendelian Randomization

Reviewer #2: Social science genomics

Reviewer #3: Mendelian Randomization

REVIEWER COMMENTS:

Reviewer #1:

Remarks to the Author:

General comments

The genetic instruments (for the main exposure and the ones used in adjustment) need better description. How many variants were used? Were any removed due to close association with other variables? Were there any variants in common (or in close LD) to the various instruments? I'm left wondering whether, by adjusting for PGI for first sex, education, etc, you've just knocked out the power of your instrument. It would be interesting to do sensitivity analyses adjusting for each of these PGIs in turn, to see if one of them is primarily responsible for the attenuation. I'm also struggling to understand the implications of adjusting for the PGI of a closely related phenotype – can you cite any studies justifying this approach? Would multivariable MR (Sanderson et al 2019, IJE 48:713-727) be a better method to use?

I find the results of the study to be greatly over-interpreted, given that they rely mainly on an imprecise MR estimate (model 4) not reaching the conventional threshold of statistical significance achieved by a more precise OLS estimate (model 2). The title correctly states “no evidence”, but if you want to use this as evidence against conventional OLS estimates, you need to test whether this null MR result is compatible with the analogous conventional estimate. Any interpretation should also ask why the unadjusted MR estimate (model 3) is so amplified relative to both its analogous OLS estimate (model 1) and the adjusted MR estimate (model 4).

Detailed comments

Abstract: “...to identify the causal effects of maternal and paternal ages at first birth on children’s test scores...”. I think you need to specify right from the start that it’s firstborn children only. Otherwise it appears that you’re examining age at first birth and outcomes in all children, which has rather different implications.

Introduction: “...The interpretation of these estimates as causal has been questioned because of the possibility of unobserved confounding variables...”. Also, many fixed effect regression models adjust for offspring DOB which risks spurious results because it is perfectly co-linear with parental age within a family. Correctly omitting this variable means that the estimated effect of parental age includes the causal but dull process whereby delaying parenthood can affect the child simply because the child is born into a later world (Holt 2014 *Jama Psychiatry* 71:432-438 / Carslake et al 2017 *Scientific Reports* 7:45278).

Introduction: “We therefore control for the PGIs of heritable AFB predictors...”. Expand PGI at first use. Also, I’m guessing these PGIs were from the parent, but this should be specified (also in the methods).

Introduction: “We control for the PGI of partner’s AFB to take assortative mating into account”. This would at least partially control for AM in the parental generation but I think AM in the grandparents could also bias estimates if it means that in each parent, the transmitted and non-transmitted genotypes are non-independent due to excess homozygosity. You could perhaps test if variants are in HWE to test this?

Introduction: “In our view, violations of the assumptions underlying the IV approach would most likely inflate the causal estimates. In other words, possible violations of the independence assumption or the exclusion restriction are likely to lead to deviations from zero and are unlikely to bias estimates downwards or to balance each other out”. Why? This opinion needs some justification. I think it also belongs in the discussion.

Introduction: Overall, the introduction feels too long, going into some detail which should be in the methods or discussion.

Table 1: The title should state that these are firstborn children.

Table 1: Rather than just controls yes/no, the figure legend should describe the adjustment set used in each model (were models 1 and 3 adjusted for nothing at all?). Are the figures in brackets SE? This should be stated (though see comment about reporting CI). If all models had the same N, it can just be reported in the legend instead of in the table.

Table 1: It would be more informative to give 95% CI than SE (if that's what is in the brackets). And if you really want to have P-values, please report the actual values rather than just asterisked categories. In terms of instrument strength, it's good to report R-squared as well as F statistics, and I think in adjusted models these should be partial values for the exposure of interest.

Table 1: Either in the table or in its legend, you need to report the units of the estimates – are they SD of test score per additional year of AFB?

Tables: I'd also like to see a table of the PGI-exposure and PGI-outcome scores reported separately, and perhaps a table of associations between the PGI and measured covariates.

Results: "...mother's AFB is associated with a 2.8% of an SD increase in test scores...". State that this is an association per year of mother's AFB (I presume).

Results: "The IV models without our control variables—including genetic pleiotropy...". This suggests that pleiotropy, stratification and AM are the control variables, rather than the processes you are hoping to account for by controlling – please re-phrase.

Results: "The estimates for both maternal and paternal AFB become negative and are no longer statistically significant". While this isn't technically untrue, I think it misses the point. The adjusted MR estimates could not be differentiated from the null, but they were so imprecise that they don't rule out quite a large positive association. In particular, you need to ask whether the MR estimates can be differentiated from the analogous OLS estimates - you could test this using a Durbin-Wu-Hausman test.

Discussion: "Our findings contradict sociological expectations of the positive effects of higher maternal and paternal ages on child education". Given the low power, I think this is over-stating the case. Also later on "However, our results demonstrate that these factors do not influence children's test scores".

Discussion: "Biologists observed potentially detrimental mental health consequences of aging...". These results need references. Also for "The MR approach in social sciences has been promoted and challenged".

Methods: There's a lack of information about exclusions and sample sizes. For example, were the MoBa data restricted to complete trios? If there were other reasons for exclusion I suggest a full flow chart, perhaps in the appendix.

Methods: "Results for each subject are available in the Online Appendix and are fully in line with those reported here". The fact that you did these sensitivity analyses should be in the analyses section of the methods and the result that they were similar to the main analysis should be in the results.

Methods: "We standardize the test scores within each subject and birth year". Please clarify – do you mean to mean 0 and SD 1?

Methods: How many variants were used to create the PGI? Were any variants removed due to association with other phenotypes? The appendix should include details of all genetic instruments.

Methods: Please report how transmitted and non-transmitted alleles (I think it's more correct to say alleles than genes here) were distinguished.

Methods: You refer to numbered models in the results table, so the description of adjustment should define these numbered models precisely. There's a lot of information missing in the description of the models. Were the same variables adjusted for in the gene-exposure and gene-outcome models used in each MR? Was birth year treated as continuous or categorical? The methods should include all the analyses performed, but the unadjusted analyses, the sensitivity analyses on the test score components and the OLS analyses are not mentioned.

Figure 1: I applaud the inclusion of a DAG, but I found this one hard to follow. The DAG should represent the assumed causal relationships between variables (observed or unobserved) on which you designed your analysis, but this one (and its legend) look as if they're trying to describe the analysis. For example, nodes should be labelled with what the variables are, not with the processes those variables are intended to adjust for. The subscripts are not really explained. I think the DAG should include the transmitted alleles and the genotype of the offspring and partner, to illustrate how the analysis of non-transmitted alleles is supposed to isolate the genetic nurture effects.

Appendix: Table S1 is never referred to in the text – why was it done? The analysis should be mentioned in the methods and the results described in the results section (probably very briefly in both cases).

Appendix: You seem to have slightly different sample sizes for each subject. I'm guessing this was because of missing data. Your methods should explain how you dealt with this.

Reviewer #2:

Remarks to the Author:

This delightful study pairs the right statistical technique with the right data to address a relevant research question. It is a shining example of Mendelian randomization (MR) done right. Aside from relevance, assumptions underlying instrumental variable

estimation cannot be formally tested. By leveraging data on genetic trios, the Authors convincingly address all common concerns about violations of MR conditions. Even distant dynastic effects are accounted for by the inclusion of grandparental education.

1. I consider it a missed opportunity that Table 1 only reports the naïve IV model, which does not control for genetic pleiotropy, population stratification or assortative mating, and the preferred IV model, which controls for all three. Most researchers will not have such rich data at their disposal, and learning which controls derived from a gold standard dataset of genotyped trios primarily contribute to eliminating the association between age at first birth and child test scores could be very informative. Suppose we only have a mother-child pair genotyped and cannot control for assortative mating. What would the estimates look like? Or we only have genotyped parents and cannot determine which genes were not inherited? Or we only have the genotype of one parent? I'm suggesting reporting some intermediate MR models reflecting best practice when constrained by lesser data. I also question not including the ten genetic principal components in the naïve model. I think at this point it has become standard practice.

2. I think that the Authors are a bit quick to interpret their findings as contradicting the sociological expectations of positive effects of higher maternal and paternal ages on child education. Consideration should be given to the unique circumstances of Norway. Over the study period, Norway enjoyed one of the highest incomes per capita among OECD countries (<https://data.oecd.org/chart/7dJQ>) coupled with very low income inequality as measured by the Gini coefficient (<https://data.oecd.org/chart/7dJS>). On top of that, Norway provides a generous child benefit until age 18. Given all of this, resources available to parents in Norway may not change with age to the same extent that they do in countries on which the sociological studies were based, thus lowering the socioeconomic premium for postponing parenthood.

3. The fact that the data comes from a national registry is a clear benefit and eliminates concerns about the findings not being representative of the Norwegian population. However, I don't believe that this affects ascertainment bias, as it is introduced at the GWAS stage. The source data for the GWAS used in the manuscript largely came from the UK Biobank, which had a low participation rate and is known not to be a perfect reflection of the general population.

4. The disadvantage of multivariable MR is that it leads to wide confidence intervals, especially when controlling for multiple pleiotropic pathways. I wonder if using Genomic SEM to subtract polygenic signals from the 5 PGIs that the current model conditions on would improve precision of the estimates.

Minor point:

--On page 6 there is a typo -- "social medication"

Reviewer #3:

Remarks to the Author:

The study by Grätz et al, investigates whether parental age at first birth is causally influencing offspring educational attainment using the principles of MR. This could be a very interesting study, which unfortunately at the current stage is quite difficult to review due to the absence of information/details on the methods. Specifically:

A. In the introduction section, the authors refer to the approaches as advanced MR, isn't this approach a within-families MR study? Or at least based on the principles of within-families MR?

B. Furthermore, in the introduction the authors suggest a number of potentially interesting analyses to minimise bias and capture the causal relationships of interest. However, when I moved to the methods section there was no detail on how e.g., the IVs were created- particularly the ones using un-transmitted maternal/ paternal alleles. Whether they adjusted for offspring genotype, what was the sample size, did they use MoBa trios or dyads or both? The above are some examples. I would strongly recommend adding further details for all the analyses conducted in the manuscript.

C. Although at this stage I cannot have a clear view on the work, considering that methodological detail is missing, I think that conceptually could be interesting. The authors might want to consider the additions of: 1. an observational analysis, 2. a two-sample MR 3. adding in the models the offspring genotype that might influence the offspring's own academic outcomes.

Version 1:

Decision Letter:

10th July 2024

Dear Prof Grätz,

Thank you once again for your revised manuscript, entitled "No Evidence for Positive Causal Effects of Maternal and Paternal Ages at First Birth on Children's Test Scores at Age 10," and for your patience during the re-review process.

Your manuscript has now been evaluated by two of the reviewers who evaluated your original manuscript in the previous round of

review. All reviewer feedback is included at the end of this letter. Although the reviewers found your manuscript to have improved during revision, they also raise some important outstanding concerns. We remain interested in the possibility of publishing your study in Nature Human Behaviour, but would like to consider your response to these outstanding concerns in the form of a revised manuscript before we make a decision on publication.

In sum, we invite you to revise your manuscript taking into account all reviewer and editor comments. We are committed to providing a fair and constructive peer-review process. Do not hesitate to contact us if there are specific requests from the reviewers that you believe are technically impossible or unlikely to yield a meaningful outcome.

We hope to receive your revised manuscript within 4-8 weeks. I would be grateful if you could contact us as soon as possible if you foresee difficulties with meeting this target resubmission date.

- Include a "Response to the editors and reviewers" document detailing, point-by-point, how you addressed each editor and referee comment. If no action was taken to address a point, you must provide a compelling argument. This response will be used by the editors and reviewers to evaluate your revision.
- Highlight all changes made to your manuscript or provide us with a version that tracks changes.

Link Redacted

We look forward to seeing the revised manuscript and thank you for the opportunity to review your work. Please do not hesitate to contact me if you have any questions or would like to discuss these revisions further.

Sincerely,

Giacomo Ariani, PhD
Senior Editor
Nature Human Behaviour

Reviewer expertise:

Reviewer #1: Mendelian Randomization

Reviewer #2: Social science genomics

Reviewer #3: Mendelian Randomization

REVIEWER COMMENTS:

Reviewer #1:

Remarks to the Author:

General comments

Thank you for the extensive improvements to the manuscript. This is a valuable study which should be published, but I do have a few remaining concerns.

As you describe in the manuscript (P4L77-83), MR can be biased by pleiotropy. You could mention that the further downstream the exposure is from the immediate effects of the variants, the greater the risk of pleiotropy involving traits which mediate the genetic effect on the exposure. An exposure like AFB must therefore be particularly vulnerable to this bias. Much of the interest of this work is in the illustration of the measures you have taken to identify and reduce this bias, but it should be recognised that the PGIs you adjust for are unlikely to account for all of the bias from pleiotropy. I also wonder if the adjustment for the education PGI, in particular, could introduce some new biases (see below).

I think it is very important for the reader to be told how correlated the various PGI are, and whether they have any SNPs in common (see comment on Results P6L132-133). My concern is that adjusting for the education PGI will induce correlation among the SNPs which contribute to it. If some of these SNPs also contribute to the AFB PGI (or are in close LD, which would make the PGIs correlated), then the AFB PGI will be associated with all SNPs contributing to the education PGI. SNPs contributing to the education PGI clearly have an effect on the outcome (offspring education) if they are inherited (have you tried adjusting for

education PGI in the offspring?), so while this step blocks some potential violations of the IV assumptions, it also opens others. You have focussed (wisely, I think) on the lack of a clear association in models 6-12 (as numbered in Fig 2) rather than the negative association in model 13. In that case, though, some of your conclusions still feel as if they go beyond what your results justify, given that the OLS results lie well within the CI of IV models 6-12. On P8L179-181 you write “our results are inconsistent with these expectations. Our findings suggest that aging itself does not positively impact children’s test scores”. An imprecise estimate is not inconsistent with a more precise estimate which falls well within the former’s CI. On P10L215 you claim that your results “contradict the idea the parental ages lead to better children’s education” – this too should be toned down.

Detailed comments

Introduction P5L100-102: Thanks for adding the HWE check, but I think the wording is confusing – it could be read to mean that you removed SNPs which WERE in HWE. I presume what you mean is that you removed SNPs which were out of HWE, at a threshold of $P < 1 \times 10^{-6}$. Please can you re-phrase to make this clear and state that the figure is a threshold P value? Also, it would be interesting to know how many SNPs, otherwise suitable for inclusion in the PGI, were excluded in this manner. Could you report this briefly in the Results?

Introduction P5L102: I think the word “as” should be omitted from “In addition, as we control for the respective PGI of inherited SNPs...”

Figure 1: This is much better, thank you. The font sizes and arrow heads are all a bit small to be read easily – could you tidy up the presentation of the DAG?

Figure 1: You have an arrow from “Other PGIs” to “PGI Age at first birth”, but I don’t think one set of variants can be said to cause another within the same person. Rather, I would say they have a common cause (broadly – who your parents are). This is potentially important when interpreting your final model.

Figure 1: In the notes, please list what the “Other PGIs” are. Also, please explain what the different colours of the nodes and edges are indicating.

Results P6L125: In your revised analysis, the attenuation of the OLS estimates on adjusting for control variables should no longer be described as “slight” – from 0.037 SD to 0.014 and from 0.023 SD to 0.009 is quite a big reduction, at least on a relative scale.

Results P6L131: The estimate for the unadjusted OLS model in mothers is 12.6% in Table 1 and 13.6% here – please check.

Results P6L130-132: These are huge estimates – The paternal age result suggests that the educational attainment of the child of a 40-year-old father would be four standard deviations higher than for the child of a 20-year-old father! I think it would be worth commenting on this (implausible?) magnitude in the Discussion and considering why they are so much higher than the OLS estimates, if you have any suggestions.

Results P6L132-133: Please can you report correlations among the control variables and between control variables and AFB PGI in parents? It would be particularly helpful to know how the AFB PGI is associated with the other PGIs. This could be a correlation matrix in the supplement.

Results P7 L142: Please specify that the influential inherited genes here are specifically the genes included in the educational attainment PGI.

Table 1: I think there’s a typo in the LCL for the maximally adjusted OLS estimate in mothers – a missing zero? There’s also the inconsistency between text and table mentioned above for the unadjusted OLS model in mothers.

Table 1: It’s helpful to have the models numbered like this, but please can you make the numbers match those in Figure 1 (even though that would leave gaps in the numbers in Table 1)? This would make it much easier to see how they correspond.

Table 1: I think it would be better to refer to it as an F statistic (or just F), and the notes should say that it’s an F stat for the effect of the instrument on the exposure. In fact, in the adjusted models, this should be the `_partial_` F statistic (for the instrument). If this is what you report, please state so.

Table 1: The BIC here don’t match those for either component model in the supplement, so am I right in thinking they are for the combined IV model? The notes should clarify this. Also, I suspect BIC are only comparable within a model type (i.e. you can’t compare an IV model with an OLS one) but I’m not sure of this – do you have any evidence one way or the other? If it is the case, it should be noted.

Table 1. Thanks for stating the units of the outcome, but the units of the estimates also require units of the exposure, so it would be better to say that the units of the estimates are SD of educational outcome per year of AFB. Units of age might seem obvious, but the reader might wonder if they too are SD.

Table 1: You could save space and make the table easier to read by extending “Maternal (/Paternal) age at first birth” across the columns. You could also add the sample size for each parent to this text to save using a whole row for a single value below.

Figure 2: This graph is really helpful in visualising the results, but my immediate response was to sketch the OLS estimates on for comparison – please can you add them?

Figure 2: As mentioned previously, consistent model numbers between Figure 2 and Table 1 would make interpretation easier. It might also help if you used different symbols to distinguish OLS and IV models.

Figure 2: Having different scales on the Y axis for mothers and fathers gives a false impression that the estimates are very similar in each parent. Please can you make the scales match?

Figure 2: In the Y axis titles, please give the units of the estimates (SD of educational attainment per year of AFB).

Tables S1 & S2: Several of my comments on Table 1 also apply to the supplementary tables.

Tables S1 & S2: There’s nothing in the “F-test” rows. If the values in Table 1 are partial F stats for the instrument in predicting the exposure, then I think you could omit these rows from the supplement (or repeat the values in Table S1 only)

Tables S1 & S2: Please state what the units of the estimates are. In particular, how are the PGIs scaled?

Discussion P7L158-160: Elsewhere (P7L146-148, P2L32-34), you stop short of claiming a negative effect of AFB on offspring educational attainment and emphasise the lack of a positive effect in IV models accounting for direct genetic effects. I think this is wise given the possible complications around adjusting for the education PGI. But if we consider the IV models accounting for direct genetic effects to be null, then this suggests only that non-transmitted parental genetics have a null effect on the outcome (not negative). You didn’t add the non-transmitted effect to the transmitted effect (cancelling it out), you removed the transmitted effect from the combined effects. In other words, I don’t think you can say that the non-transmitted genetics have an “opposite” effect to the transmitted genetics because the opposite to a positive effect would be a negative effect.

Discussion P8L175-177: A small point, but an effect of parental wealth could be non-causal in terms of AFB as suggested here (lifelong wealth and AFB are both caused by SEP) but it could also be mediating a causal effect (individuals get richer as they get older).

Discussion P8L182: I find it hard to extract the intended meaning from “Our findings are rather but also not robustly...”, can you re-

phrase?

Discussion P10L215: should be "contradicts the idea that" (not "the") – but see my objections to this statement.

Methods P11L248: There should be no apostrophe after "parents".

Methods P11L258-259: "our polygenic indices (PGIs) are based on all these SNPs" – presumably you mean that all these SNPs were candidates for the PGIs, and then those associated with each trait in the discovery sample were used to make the PGI? The current wording could be taken to mean that all 6.9 million SNPs actually contributed numerically to the PGIs – can you clarify this please? Also, was Mills et al 2021 the discovery study used for the control PGIs as well as for the ADB PGI? I think the text hints at this but it's not stated explicitly.

Methods P12 L262-274: The OLS analysis should be mentioned somewhere in this section.

References P14L331: The Mills 2021 citation has a stray "262" in the title and I think the page numbers are probably from an online advance version rather than the final version (which is pp1717-1730).

Reviewer #2:

Remarks to the Author:

All of my original comments were adequately addressed. As the manuscript was extensively revised, I have these new comments to share:

1. The increase in the sample size is not discussed in the rebuttal. Both versions of the manuscript state that "Our sample is restricted to first-born children who were born between 2001 and 2008 in Norway", yet the sample size has gone up by over 50%. In re-reading the manuscript it would have been beneficial to understand where these new cases came from and how they could be different from those originally used, especially with the estimates in the full model now significantly negative.
2. The estimation strategy was changed from using only non-transmitted alleles in making the parental AFB PGI to using the full PGI and controlling for the child's PGI for AFB. In the rebuttal the Authors comment that this change allowed them to disentangle the effects of genetic inheritance from other processes. This discussion has not made it into the manuscript though. I think that the rationale for this modeling choice should be discussed in the manuscript, with a robustness check using the non-transmitted alleles strategy presented in the appendix.
3. I found myself wanting for commentary on what distinguishes the PGI for education from the other PGI controls. My guess was that it captured cognitive performance, but then I reached for the AFB GWAS paper (Mills et al. 2021). They report that cognitive performance was not a mediating trait in the genetic correlation of AFB with education and further note that "the genetic correlation between years of education and AFB is largely a product of a strong bidirectional relationship between these traits, rather than being both downstream of a common identified cause". In light of the full model tipping negative after the inclusion of this variable, some discussion of why it is a particularly potent control for pleiotropy is warranted.
4. It would be helpful to have a legend for the DAG in Figure 1 that defines colors for nodes and edges.
5. I liked how the note to Figure 1 in the original manuscript grouped model variables by processes that they are meant to control for. I think that this is somewhat lost in the revised manuscript. Perhaps color-coding the bars in Figure 2 into categories of population stratification, genetic inheritance, assortative mating, and pleiotropy would bring back that information.
6. "one year later mother's AFB" sounds awkward. Maybe "one year delay in mother's AFB"? Also "with higher performance by 2.3%" could be revised to "with performance rising by 2.3%".
7. In their rebuttal the Authors write that they updated the title and abstract to indicate that the analysis is restricted to firstborn children. These revisions are missing from the June 7 version of the manuscript shared with reviewers. Similarly, the reference to (Nivard et al. 2024) invoked in response to comment 23 does not appear in the manuscript.

Version 2:

Decision Letter:

Our ref: NATHUMBEHAV-23103315B

22nd October 2024

Dear Dr Grätz,

Thank you for submitting your revised manuscript "No Evidence of Positive Causal Effects of Maternal and Paternal Age at First Birth on Children's Test Scores at Age 10" (NATHUMBEHAV-23103315B). It has now been seen by one of the original referees and their comments are below. As you can see, the reviewer finds that the paper has improved in revision. We will therefore be happy in principle to publish it in Nature Human Behaviour, pending minor revisions to satisfy the referees' final requests and to comply with our editorial and formatting guidelines.

We are now performing detailed checks on your paper and will send you a checklist detailing our editorial and formatting requirements within two weeks. Please do not upload the final materials and make any revisions until you receive this additional information from us.

Sincerely,

Giacomo Ariani, PhD
Senior Editor
Nature Human Behaviour

Reviewer #1 (Remarks to the Author):

Thank you for your comprehensive responses to my comments. I have no major comments on the revised manuscript but spotted a few minor issues which I suggest correcting before publication.

Detailed comments

L80: Did you mean "risk of externalising behaviour" (not "or")?

L102: This threshold would normally be written for publication with the exponent superscripted.

L125: Because you define "controls" in the table as excluding parental PGI for education, I think you need to write "adding the control variables and parental PGI for education" or at least "adding all the control variables" to describe model 13.

Figure 2: The last bit of the figure title doesn't quite sound right. Maybe "...different model specifications with covariates added stepwise".

L144: Similarly to the comment above, I think "...performing analyses with covariates added stepwise" would be better.

Figure 2: Are the CI for the OLS estimates just so small that they don't show up beyond the thickness of the point? If so, I think this should be mentioned in the table footnote just in case a reader thinks they have been omitted.

Table S2: The title is missing the word "age" in "PGIs for at first birth"

L245: Can you specify which analysis this power calculation is referring to - the OLS or the IV?

Version 3:

Decision Letter:

Dear Prof Grätz,

We are pleased to inform you that your Article "No Evidence of Positive Causal Effects of Maternal and Paternal Age at First Birth on Children's Test Scores at Age 10", has now been accepted for publication in Nature Human Behaviour.

With best regards,

Giacomo Ariani, PhD
Senior Editor
Nature Human Behaviour

P.S. Click on the following link if you would like to recommend Nature Human Behaviour to your librarian
<http://www.nature.com/subscriptions/recommend.html#forms>

** Visit the Springer Nature Editorial and Publishing website at http://editorial-jobs.springernature.com?utm_source=ejp_NHumB_email&utm_medium=ejp_NHumB_email&utm_campaign=ejp_NHumB for more information about our career opportunities. If you have any questions please click [here](mailto:editorial.publishing.jobs@springernature.com).

No Evidence for Positive Causal Effects of Maternal and Paternal Ages at First Birth on Children's Test Scores at Age 10. Response to the Reviewers.

June 7, 2024

Dear Reviewers,

thank you very much for the productive criticism of our work and for providing us with an opportunity to submit a revised version of our manuscript. In fact, the comments were very fair and helpful; we are convinced that implementing them improved considerably on the quality of our manuscript. Please find below a detailed explanation of how we addressed the comments in the revised version of the manuscript.

Kind regards,

The Authors

Reviewer 1

1. The genetic instruments (for the main exposure and the ones used in adjustment) need better description. How many variants were used? Were any removed due to close association with other variables? Were there any variants in common (or in close LD) to the various instruments? I'm left wondering whether, by adjusting for PGI for first sex, education, etc, you've just knocked out the power of your instrument. It would be interesting to do sensitivity analyses adjusting for each of these PGIs in turn, to see if one of them is primarily responsible for the attenuation. I'm also struggling to understand the implications of adjusting for the PGI of a closely related phenotype – can you cite any

studies justifying this approach? Would multivariable MR (Sanderson et al 2019, IJE 48:713-727) be a better method to use?

We thank the reviewer for pointing out these very insightful questions and address them in the following one by one: a) We agree that a description of the genetic data is required. We added the following description to the revised version of the manuscript: “In this study, we use data from MoBa (Corfield et al. 2022; Magnus et al. 2016), which is a population-based cohort study conducted by the Norwegian Institute of Public Health. Participants were recruited in Norway from 1999 to 2008. Pregnant women consented to participate in 41% of cases. The cohort includes approximately 114,500 children, 95,200 mothers and 75,200 fathers. The current study is based on version 12 of the quality-assured data files released for research in 2019. The establishment of MoBa and the initial data collection were based on a license from the Norwegian Data Protection Agency and approval from the Regional Committees for Medical and Health Research Ethics. The MoBa cohort is regulated by the Norwegian Health Registry Act. Our sample is restricted to first-born children who were born between 2001 and 2008 in Norway to focus on the causal effects of maternal and paternal ages at first birth.” (Page 10) b) No variables were removed due to LD or association with the instrument. Instead, we statistically control for potentially shared effects with other traits across the whole genome by including polygenic indices for other important traits. We describe this in more detail in the revised version of the manuscript on page 4: “Pleiotropy leads to a violation of the exclusion restriction assumption central to the IV approach—that the instrument does not influence the outcome other than through the instrumented variable. We therefore control for the Polygenic Indices (PGIs) of heritable AFB predictors in the parental generation, specifically sexual, contraceptive, and smoking behavior, attention deficit hyperactivity disorder, and educational attainment (Mills et al. 2021). “ And on pages 11-12: “We identify AFB based on non-transmitted alleles of the focal parent by controlling for inherited AFB alleles in a PGI to avoid

bias due to genetic inheritance (Nivard et al, 2024). We furthermore consider population stratification controlling for grandparental education and the first ten genetic principal components, AM controlling for the partner's PGI, demographic characteristics (birth year and sex of the child) and control for the PGIs for age at first sexual intercourse, age at smoking initiation, age at first use of oral contraceptives, attention deficit hyperactivity disorder, and educational attainment, which have previously been linked to the AFB PGI (Mills et al. 2021) and may influence parental nurturing behavior relevant to child test scores.” Regarding c) and d): Following the reviewers' suggestions, we added a more detailed depiction of our analyses in a new specification plot (Figure 2) in the main text, stepwise including all covariates. Statistical power decreases indeed, especially when controlling for parental PGI for education. However, also estimates change substantially and we consider the inclusion of these controls as meaningful and crucial. e) To our knowledge, this is a novel approach to take pleiotropy into account and is shown as being effective, in particular highlighting the role of parental PGI for educational attainment as a confounding of the AFB-children's score link. In principle, the logic of including correlated PGIs as control is known from genetic nurture studies, in the same fashion we use those scores to identify non-inherited genetic effects (e. g. Nivard, Michel G., et al. More than nature and nurture, indirect genetic effects on children's academic achievement are consequences of dynastic social processes. *Nature Human Behaviour* (2024): 1-8.) f) Thanks for your suggestion regarding multivariate MR proposed in the very insightful study by Sanderson et al 2019. However, whilst Sanderson et al.'s method would be a great alternative approach, it requires exposure data on the control variables which we have not phenotypically present in our data. We therefore have to stick with the approach currently employed.

2. I find the results of the study to be greatly over-interpreted, given that they rely mainly on an imprecise MR estimate (model 4) not reaching the conventional threshold of

statistical significance achieved by a more precise OLS estimate (model 2). The title correctly states “no evidence”, but if you want to use this as evidence against conventional OLS estimates, you need to test whether this null MR result is compatible with the analogous conventional estimate. Any interpretation should also ask why the unadjusted MR estimate (model 3) is so amplified relative to both its analogous OLS estimate (model 1) and the adjusted MR estimate (model 4).

We thank the reviewer for these comments. We phrased the entire manuscript in a way that we find no evidence for effects of parental ages whilst it is difficult to prove the null hypothesis of no effects because of the large confidence intervals. We entirely agree with the reviewer that it is crucial to unpack our modelling strategy empirically. We therefore added a model specification analysis and the following more detailed descriptions of our findings: “The first stage F-statistics of the IV models are substantial and our analyses do not suffer from weak instrument bias. The IV models without our control variables show a strong inflation of estimates compared with the OLS models for both maternal (13.6%) and paternal (19.8%) ages at first birth (Models 5). However, once we enter the control variables into the IV models, effects become statistically insignificantly different from 0 (Models 7). Once we add a control for the parental educational attainment PGI the estimates turn around and we find, in the full model, even evidence for a negative causal effect of AFB on test scores at age 10: The estimates for both maternal and paternal AFB become statistically significantly negative (mothers: – 8.8%; fathers: –19.3%) (Models 8).

To unpack these findings, we analyse our model specifications in detail, conducting a stepwise analysis including the covariates under study (see Figure 2). We can identify two major shifts in estimates of our IV across specifications. First, the full PGI of AFB for both parents are positively predictive of children’s test scores when combining both indirect genetic effects and inherited alleles. Once the model controls for inherited genes of the child,

identifying nurturing or ancestral effects for the instrument respectively, the estimate approaches zero. Second, including the parental PGI for educational attainment turns estimates negative.

In summary, we arrive at the conservative conclusion that there is no robust evidence for positive effects of paternal and maternal ages at first birth on child test scores at age 10 in Norway.” (Pages 6-7)

3. Abstract: “...to identify the causal effects of maternal and paternal ages at first birth on children’s test scores...”. I think you need to specify right from the start that it’s firstborn children only. Otherwise it appears that you’re examining age at first birth and outcomes in all children, which has rather different implications.

We agree and updated the text accordingly.

4. Introduction: “...The interpretation of these estimates as causal has been questioned because of the possibility of unobserved confounding variables...”. Also, many fixed effect regression models adjust for offspring DOB which risks spurious results because it is perfectly co-linear with parental age within a family. Correctly omitting this variable means that the estimated effect of parental age includes the causal but dull process whereby delaying parenthood can affect the child simply because the child is born into a later world (Holt 2014 Jama Psychiatry 71:432-438 / Carslake et al 2017 Scientific Reports 7:45278).

Thank you for pointing this out. We added the argument and the references to the introduction: “In addition, it is unclear whether one should adjust for the offspring’s year of birth. Adjusting for offspring’s year of birth is problematic because it is in a family fixed effect model perfectly collinear with parental ages. However, not adjusting for year of birth to avoid the

multicollinearity issue results in any parental age effect being driven possibly by the difference in the time period the different siblings are born in (Carslake et al. 2017; Holt 2014).“ (Pages 3-4)

5. Introduction: “We therefore control for the PGIs of heritable AFB predictors...”. Expand PGI at first use. Also, I’m guessing these PGIs were from the parent, but this should be specified (also in the methods).

They were from the parent. We thank the reviewer for this comment and adapted the sentence to: “We therefore control for the Polygenic Indices (PGIs) of heritable AFB predictors in the parental generation, specifically sexual, contraceptive, and smoking behavior, attention deficit hyperactivity disorder, and educational attainment (Mills et al. 2021).” (Page 4)

6. Introduction: “We control for the PGI of partner’s AFB to take assortative mating into account”. This would at least partially control for AM in the parental generation but I think AM in the grandparents could also bias estimates if it means that in each parent, the transmitted and non-transmitted genotypes are non-independent due to excess homozygosity. You could perhaps test if variants are in HWE to test this?

We thank the reviewer for this comment and agree that grandparental assortative mating can still influence our results. We followed the suggested strategy to take this into account and QCed variants in Hardy-Weinberg equilibrium 1×10^{-6} in the parental generation. Second, we control for inherited genes with the respective PGI to take the effect of potentially remaining excess homozygosity into account. We state our strategy in the manuscript as follows: “To take grandparental assortative mating into account leading to potential excess homozygosity, we remove single nucleotide polymorphisms (SNP) in Hardy-Weinberg equilibrium of 1×10^{-6} (Corfield et al. 2022).” (Page 5)

7. Introduction: “In our view, violations of the assumptions underlying the IV approach would most likely inflate the causal estimates. In other words, possible violations of the independence assumption or the exclusion restriction are likely to lead to deviations from zero and are unlikely to bias estimates downwards or to balance each other out”. Why? This opinion needs some justification. I think it also belongs in the discussion.

Thanks for highlighting this. We deleted this statement. In fact, the deviations could go in any direction. Therefore, we prefer not to make any statement in any direction.

8. Introduction: Overall, the introduction feels too long, going into some detail which should be in the methods or discussion.

We agree and followed the reviewer’s advice. We shortened the introduction and moved some details to the methods and discussion section.

9. Table 1: The title should state that these are firstborn children.

We updated the title to “No Evidence for Positive Causal Effects of Maternal and Paternal Ages at First Birth on Children’s Test Scores at Age 10” making thus clear that these are firstborn children.

10. Table 1: Rather than just controls yes/no, the figure legend should describe the adjustment set used in each model (were models 1 and 3 adjusted for nothing at all?). Are the figures in brackets SE? This should be stated (though see comment about reporting CI). If all models had the same N, it can just be reported in the legend instead of in the table.

The revised version of Table 1 includes this information. We now report confidence intervals. The N of the models were different for mother and fathers due to missing values. Please see also the new Figure 2 in the revised version of the manuscript, which includes controls stepwise.

11. Table 1: It would be more informative to give 95% CI than SE (if that's what is in the brackets). And if you really want to have P-values, please report the actual values rather than just asterisked categories. In terms of instrument strength, it's good to report R-squared as well as F statistics, and I think in adjusted models these should be partial values for the exposure of interest.

We agree and now report confidence intervals in Table 1.

12. Table 1: Either in the table or in its legend, you need to report the units of the estimates – are they SD of test score per additional year of AFB?

These are SD per additional year. We added the information to Table 1.

13. Tables: I'd also like to see a table of the PGI-exposure and PGI-outcome scores reported separately, and perhaps a table of associations between the PGI and measured covariates

We report these models now as two tables in the new *Online Supplement*.

14. Results: "...mother's AFB is associated with a 2.8% of an SD increase in test scores...". State that this is an association per year of mother's AFB (I presume).

Thanks for pointing this out. We now adapted the text considering our new results to: "The OLS models, in line with earlier research, show positive associations between both maternal and paternal ages at first birth and child education. Without control variables, one year later

mother's AFB is associated with a 3.7% of an SD increase in test scores in fifth grade (around age 10) (Model 1)." (Page 6)

15. Results: "The IV models without our control variables—including genetic pleiotropy...". This suggests that pleiotropy, stratification and AM are the control variables, rather than the processes you are hoping to account for by controlling – please re-phrase

We agree with the reviewer's comments and drop the details about the control variables in the sentence. Please note that we also modified the DAG in Figure 1 in order to clarify that our control variables aim to approximate these phenomena instead of measuring them.

16. Results: "The estimates for both maternal and paternal AFB become negative and are no longer statistically significant". While this isn't technically untrue, I think it misses the point. The adjusted MR estimates could not be differentiated from the null, but they were so imprecise that they don't rule out quite a large positive association. In particular, you need to ask whether the MR estimates can be differentiated from the analogous OLS estimates - you could test this using a Durbin-Wu-Hausman test.

We now report, as suggested above confidence intervals and we are careful in phrasing that we find no evidence in favor of positive age effects. In other words, we are clear about the result that the confidence intervals are large.

17. Discussion: "Our findings contradict sociological expectations of the positive effects of higher maternal and paternal ages on child education". Given the low power, I think this is over-stating the case. Also later on "However, our results demonstrate that these factors do not influence children's test scores".

These are great points and we considering our new findings, our interpretation has been adapted accordingly: “Our findings show no support for sociological expectations of the positive effects of higher maternal and paternal ages on child education next to mechanisms linked to education. The positive association between parental aging and children’s academic performance had been indeed attributed by social scientists to higher socioeconomic status and the greater resources of older parents (Powell et al. 2006). However, in addition, it has been argued that higher parental age leads to more stable relationships and better parenting (Kalmijn and Kraaykamp 2005), which is beneficial for children’s academic performance. Our results are inconsistent with these expectations. Our findings suggest that aging itself does not positively impact children’s test scores.” (Page 8)

18. Discussion: “Biologists observed potentially detrimental mental health consequences of aging...”. These results need references. Also for “The MR approach in social sciences has been promoted and challenged”.

We thank the reviewer for pointing this out and added the references on Pages 8-9: “Biologists observed potentially detrimental mental health consequences of aging, especially related to fathers’ accumulation and inheritance of de novo mutations across the life course and increased psychiatric problems for children. Also, age-related methylation changes (Salameh et al. 2020) that may modify parenting behavior potentially influence children’s school performance substantially including episodes of psychoses (Bassett et al. 1996; MacCabe et al. 2009; McGrath et al. 1999).” And: “The MR approach in social sciences has been promoted (Benjamin et al. 2012) and challenged (Conley & Zhang 2018).” (Page 9)

19. Methods: There's a lack of information about exclusions and sample sizes. For example, were the MoBa data restricted to complete trios? If there were other reasons for exclusion I suggest a full flow chart, perhaps in the appendix.

The MoBa data was restricted to complete trios (Corfield et al. 2022). The sample sizes were 15,670 for mothers and 15,593 for fathers. There were no other reasons for exclusion.

20. Methods: "Results for each subject are available in the Online Appendix and are fully in line with those reported here". The fact that you did these sensitivity analyses should be in the analyses section of the methods and the result that they were similar to the main analysis should be in the results.

We dropped these models from the revised version of the manuscript.

21. Methods: "We standardize the test scores within each subject and birth year". Please clarify – do you mean to mean 0 and SD 1?

Thanks for pointing this out. We added the information to this sentence on page 11 of the revised version of the manuscript: "We standardize the test scores within each subject and birth year to have a mean of 0 and a standard deviation of 1."

22. Methods: How many variants were used to create the PGI? Were any variants removed due to association with other phenotypes? The appendix should include details of all genetic instruments.

We used all available variants. No variants were removed. For more information, please refer to Corfield et al. 2022.

23. Methods: Please report how transmitted and non-transmitted alleles (I think it's more correct to say alleles than genes here) were distinguished.

This is an excellent point. In our original analyses, we had constructed polygenic scores based on untransmitted alleles only. However, with our new and increased sample, we followed an alternative strategy. We utilized the full PGI of the parents and controlled for children's polygenic index to take inherited alleles into account. This also allowed us to disentangle these two processes. We added the following sentence on page 5: "We identify non-transmitted effects by jointly including parental and child's PGI for AFB in the statistical model (Bates et al. 2018; Kong et al. 2018). In this way, we minimize the possibility of observed associations between parents' and children's outcomes being due to shared instrument-related genotypes." And to the Methods Section on Pages 11-12: "We identify AFB based on non-transmitted alleles of the focal parent by controlling for inherited AFB alleles in a PGI to avoid bias due to genetic inheritance (Nivard et al, 2024)."

24. Methods: You refer to numbered models in the results table, so the description of adjustment should define these numbered models precisely. There's a lot of information missing in the description of the models. Were the same variables adjusted for in the gene-exposure and gene-outcome models used in each MR? Was birth year treated as continuous or categorical? The methods should include all the analyses performed, but the unadjusted analyses, the sensitivity analyses on the test score components and the OLS analyses are not mentioned.

Please see our new Figure 2 which adds stepwise control variables. Birth year was treated as continuous and this information is now included in Table 1.

25. Figure 1: I applaud the inclusion of a DAG, but I found this one hard to follow. The DAG should represent the assumed causal relationships between variables (observed or unobserved) on which you designed your analysis, but this one (and its legend) look as if they're trying to describe the analysis. For example, nodes should be labelled with what the variables are, not with the processes those variables are intended to adjust for. The subscripts are not really explained. I think the DAG should include the transmitted alleles and the genotype of the offspring and partner, to illustrate how the analysis of non-transmitted alleles is supposed to isolate the genetic nurture effects.

We updated the DAG for the revised version of the manuscript.

26. Appendix: Table S1 is never referred to in the text – why was it done? The analysis should be mentioned in the methods and the results described in the results section (probably very briefly in both cases).

This was a mistake. We dropped this table from the revised version of the *Online Supplement*.

27. Appendix: You seem to have slightly different sample sizes for each subject. I'm guessing this was because of missing data. Your methods should explain how you dealt with this.

We added the missing information to page 11 of the revised version of the manuscript: “We measure children’s academic performance using national standardized tests in math, reading, and English in fifth grade (around age 10). We analyze a measure averaging across subjects. In case of missing values on one subject, we build the average using the other test scores. We standardize the test scores within each subject and birth year to have a mean of 0 and a standard deviation of 1. For AFB, we include only children for whom both parents’ register as first-time parents.”

Reviewer 2

This delightful study pairs the right statistical technique with the right data to address a relevant research question. It is a shining example of Mendelian randomization (MR) done right. Aside from relevance, assumptions underlying instrumental variable estimation cannot be formally tested. By leveraging data on genetic trios, the Authors convincingly address all common concerns about violations of MR conditions. Even distant dynastic effects are accounted for by the inclusion of grandparental education.

1. I consider it a missed opportunity that Table 1 only reports the naïve IV model, which does not control for genetic pleiotropy, population stratification or assortative mating, and the preferred IV model, which controls for all three. Most researchers will not have such rich data at their disposal, and learning which controls derived from a gold standard dataset of genotyped trios primarily contribute to eliminating the association between age at first birth and child test scores could be very informative. Suppose we only have a mother-child pair genotyped and cannot control for assortative mating. What would the estimates look like? Or we only have genotyped parents and cannot determine which genes were not inherited? Or we only have the genotype of one parent? I'm suggesting reporting some intermediate MR models reflecting best practice when constrained by lesser data. I also question not including the ten genetic principal components in the naïve model. I think at this point it has become standard practice.

We thank the reviewer for their comment and agree. We therefore added Figure 2 showing all the estimates subsequently including the covariates. This is indeed very informative, and we discuss our findings in the results section, Page 7: “To unpack these findings, we analyse our model specifications in detail, conducting a stepwise analysis including the covariates under study (see Figure 2). We can identify two major shifts in estimates of our IV across

specifications. First, the full PGI of AFB for both parents are positively predictive of children's test scores when combining both indirect genetic effects and inherited alleles. Once the model controls for inherited genes of the child, identifying nurturing or ancestral effects for the instrument respectively, the estimate approaches zero. Second, including the parental PGI for educational attainment turns estimates negative." We also added the following sentences to the discussion: "Numerous studies have reported positive associations between maternal and paternal ages and children's educational outcomes even within families (Barclay and Myrskylä 2016; Duncan et al. 2018; Kalmijn and Kraaykamp 2005; Powell et al. 2006). However, it has been unclear whether these positive associations are due to underlying positive causal effects of higher maternal and paternal ages. Using an MR approach, we find no robust evidence for positive causal effects of maternal and paternal ages at first birth on children's test scores at age 10. Specification analyses unravel that the positive association emerges from a null effect, once children's genes and correlations among different PGIs are considered. Our results furthermore suggest that the PGI for non-transmitted alleles of parents' AFB has the opposite effect compared to transmitted ones as taking children's PGI for AFB into account reduces the IV estimate. However, this is likely attributable to the fact that our child AFB PGI is still including education-linked genetic effects. Our results are specific to the analysis of AFB and other research designs are needed to study the effects of parental ages at higher order births." (Pages 7-8)

2. I think that the Authors are a bit quick to interpret their findings as contradicting the sociological expectations of positive effects of higher maternal and paternal ages on child education. Consideration should be given to the unique circumstances of Norway. Over the study period, Norway enjoyed one of the highest incomes per capita among OECD countries (<https://data.oecd.org/chart/7dJQ>) coupled with very low income inequality as

measured by the Gini coefficient (<https://data.oecd.org/chart/7dJS>). On top of that, Norway provides a generous child benefit until age 18. Given all of this, resources available to parents in Norway may not change with age to the same extent that they do in countries on which the sociological studies were based, thus lowering the socioeconomic premium for postponing parenthood.

Thank you for this point. We agree and added the following paragraph to the discussion for clarification: “Our results are obtained with respect to one specific society (Norway). Norway has a high income per capita compared to other OECD countries with a low level of income inequality. In addition, the Norwegian state provides a generous amount of child benefits. For that reason, the positive causal effect of parental ages on child education may be lower in Norway than in other societies, such as the United States.” (Page 9)

3. The fact that the data comes from a national registry is a clear benefit and eliminates concerns about the findings not being representative of the Norwegian population. However, I don’t believe that this affects ascertainment bias, as it is introduced at the GWAS stage. The source data for the GWAS used in the manuscript largely came from the UK Biobank, which had a low participation rate and is known not to be a perfect reflection of the general population.

We acknowledge this limitation in the revised version of the manuscript: “Ascertainment bias may be introduced in our study because the GWAS used in our study largely came from the UK Biobank, which had a low participation rate and is known not to be a perfect reflection of the general population. However, it is unclear whether this biases our estimation with respect to the specific research question we analyze.” (Page 8)

4. The disadvantage of multivariable MR is that it leads to wide confidence intervals, especially when controlling for multiple pleiotropic pathways. I wonder if using Genomic SEM to subtract polygenic signals from the 5 PGIs that the current model conditions on would improve precision of the estimates.

We appreciate the reviewer's idea. However, we made efforts to increase power in general in our approach by increasing our sample size, which strengthens our confidence in our null findings. We opted to not conduct a genomic SEM because we consider our approach, with the stepwise addition of control variables in the new Figure 2, is more transparent.

5. Minor point: On page 6 there is a typo -- "social medication".

Thank you. We corrected this in the revised version of the manuscript.

Reviewer 3

The study by Grätz et al, investigates whether parental age at first birth is causally influencing offspring educational attainment using the principles of MR. This could be a very interesting study, which unfortunately at the current stage is quite difficult to review due to the absence of information/details on the methods. Specifically: A. In the introduction section, the authors refer to the approaches as advanced MR, isn't this approach a within-families MR study? Or at least based on the principles of within-families MR?

Unfortunately, we were not able to implement a within-family MR, which was our initial idea, due to a too low sample size. In contrast to a within-family MR, we study intergenerational relationships which are analysed between families. However, we go beyond the standard MR approach controlling for partners scores, taking genetic nurture into account, polygenic indices for other traits, etc. Nonetheless, we removed the word "advanced MR" from the introduction.

B. Furthermore, in the introduction the authors suggest a number of potentially interesting analyses to minimise bias and capture the causal relationships of interest. However, when I moved to the methods section there was no detail on how e.g., the IVs were created- particularly the ones using un-transmitted maternal/ paternal alleles. Whether they adjusted for offspring genotype, what was the sample size, did they use MoBa trios or dyads or both? The above are some examples. I would strongly recommend adding further details for all the analyses conducted in the manuscript.

Our approach changed in the revised version of the manuscript. We now use a more standard approach. We created the standard PGIs. They were not adjusted for offspring genotype. This makes our approach more transparent. The sample sizes were 15,670 for the models using maternal age at first birth and 15,593 for the models using paternal age at first birth. We only use trios.

C. Although at this stage I cannot have a clear view on the work, considering that methodological detail is missing, I think that conceptually could be interesting. The authors might want to consider the additions of: 1. an observational analysis, 2. a two-sample MR 3. adding in the models the offspring genotype that might influence the offspring's own academic outcomes.

We thank the reviewer for these suggestions and believe that we address point 1 and 3 as we consider the OLS analyses as observational study and our models control for children's PGIs. Regarding two sample MR, our understanding is that this approach is based on summary statistics. Our analytical strategy involves the regression of children's test scores on parental age at first birth. To our knowledge, no GWAS on test scores have been conducted across

generations. Please note as well that our focus on non-inherited SNPs as instrument further reduces the use of GWAS summary statistics-based MR analyses.

No Evidence of Positive Causal Effects of Maternal and Paternal Age at First Birth on Children's Test Scores at Age 10. Response to the Reviewers.

October 8, 2024

Dear Reviewers,

Thank you very much for the constructive criticism of our work and for giving us the opportunity to submit another revised version of our manuscript. Indeed, the comments were very fair and helpful; we are convinced that the implementation of these comments has significantly improved the quality of our manuscript. Below is a detailed explanation of how we have addressed the comments in the revised version of the manuscript.

Kind regards,

The Authors

Reviewer 1

Thank you for the extensive improvements to the manuscript. This is a valuable study which should be published, but I do have a few remaining concerns.

We thank the reviewer for this positive evaluation of our manuscript.

As you describe in the manuscript (P4L77-83), MR can be biased by pleiotropy. You could mention that the further downstream the exposure is from the immediate effects of the variants, the greater the risk of pleiotropy involving traits which mediate the genetic effect on the exposure. An exposure like AFB must therefore be particularly vulnerable to this bias.

We agree with the reviewer and believe this strengthens the case we make for our research design. We added the suggested remark on p. 4 of the manuscript: “This phenomenon, known as pleiotropy, can be due to shared biological influences, social mediation, or genetic effects on third common causes (Wedow et al. 2018), and can be more pronounced the further a predicted outcome is from actual biological processes-which makes AFB particularly susceptible to pleiotropy.”

Much of the interest of this work is in the illustration of the measures you have taken to identify and reduce this bias, but it should be recognised that the PGIs you adjust for are unlikely to account for all of the bias from pleiotropy.

We agree with the reviewer and added a remark in our discussion of potential remaining biases that our strategy cannot rule out remaining biases due to pleiotropy on p. 8: “While IV approaches in general, and MR approaches in particular, have been challenged for potential violations of assumptions such as the independence assumption, exclusion restriction, or population stratification (Young et al. 2019), we think it is rather unlikely that the effects of such violations perfectly balance our results towards zero or are responsible for making them negative in the full model but we cannot rule out potential biases, particularly due to pleiotropy.”

I also wonder if the adjustment for the education PGI, in particular, could introduce some new biases (see below). I think it is very important for the reader to be told how correlated the various PGI are, and whether they have any SNPs in common (see comment on Results P6L132-133). My concern is that adjusting for the education PGI will induce correlation among the SNPs which contribute to it. If some of these SNPs also contribute to the AFB PGI (or are in close LD, which would make the PGIs correlated), then the AFB PGI will

be associated with all SNPs contributing to the education PGI. SNPs contributing to the education PGI clearly have an effect on the outcome (offspring education) if they are inherited (have you tried adjusting for education PGI in the offspring?), so while this step blocks some potential violations of the IV assumptions, it also opens others.

This is always a possibility. We now provide a correlation matrix as Table S3 in the *Online Supplement* of the revised version of the manuscript. We added also these thoughts to the manuscript. First, we added the discussion to the results section: “Whilst the inclusion of the PGI of education may reduce bias by controlling for a confounder, it could also introduce overcontrol bias if the PGI for education picks up part of the effect which actually belongs to the PGI for AFB. The correlation between the PGI for AFB and the PGI for education is shown in Table S3 in the *Online Supplement*. We therefore conservatively conclude that our results provide no support to the idea that parental age positively affects children’s educational performance.” (Pages 6–7) Second, we take up the point in the discussion section: “However, at the same time, the PGI for education is a proxy measure, which captures all factors related to education, including diseases, cognitive skills, and personality traits. For this reason, conditioning on the PGI for education could also introduce overcontrol bias.” (Page 8) In summary, these reasons only further support our main point that we should look at all models in combination, and not prefer the last model 13 over the others.

You have focussed (wisely, I think) on the lack of a clear association in models 6-12 (as numbered in Fig 2) rather than the negative association in model 13. In that case, though, some of your conclusions still feel as if they go beyond what your results justify, given that the OLS results lie well within the CI of IV models 6-12. On P8L179-181 you write “our results are inconsistent with these expectations. Our findings suggest that aging itself does not positively impact children’s test scores”. An imprecise estimate is not inconsistent with

a more precise estimate which falls well within the former's CI. On P10L215 you claim that your results “contradict the idea the parental ages lead to better children's education” – this too should be toned down.

Thank you. We agree and carefully changed the wording so that it now reads: “Our results do not support sociological expectations of positive effects of higher maternal and paternal age on children's education. The positive associations between parental ages and children's academic performance have been attributed by social scientists to higher socioeconomic status and the greater resources that parents accumulate as they age (Powell et al. 2006). In addition, it has been argued that higher parental age leads to more stable relationships and better parenting (Kalmijn and Kraaykamp 2005), which is beneficial for children's academic performance. Our results do not provide evidence to support these expectations, although uncertainty remains due to the large confidence intervals.”

Detailed comments: Introduction P5L100-102: Thanks for adding the HWE check, but I think the wording is confusing – it could be read to mean that you removed SNPs which WERE in HWE. I presume what you mean is that you removed SNPs which were out of HWE, at a threshold of $P < 1 \times 10^{-6}$. Please can you re-phrase to make this clear and state that the figure is a threshold P value? Also, it would be interesting to know how many SNPs, otherwise suitable for inclusion in the PGI, were excluded in this manner. Could you report this briefly in the Results?

Thank you for pointing this out. We corrected the wording that we exclude SNPs out of HWequilibrium and we added that the threshold is a p-value threshold. Please see the revised version of the manuscript, so that it reads now on page 5: “we removed single nucleotide polymorphisms (SNP) which are out of the Hardy-Weinberg equilibrium based on a p-value threshold of 1×10^{-6} (Corfield et al. 2022)”

Introduction P5L102: I think the word “as” should be omitted from “In addition, as we control for the respective PGI of inherited SNPs...”

Thank you. We deleted the “as”.

Figure 1: This is much better, thank you. The font sizes and arrow heads are all a bit small to be read easily – could you tidy up the presentation of the DAG?

We did draw the DAG again in LaTeX.

Figure 1: You have an arrow from “Other PGIs” to “PGI Age at first birth”, but I don’t think one set of variants can be said to cause another within the same person. Rather, I would say they have a common cause (broadly – who your parents are). This is potentially important when interpreting your final model.

We agree and we deleted the arrow from the DAG.

Figure 1: In the notes, please list what the “Other PGIs” are. Also, please explain what the different colours of the nodes and edges are indicating.

The other PGIs are now listed in the notes to Figure 1. The colors were meaningless and we now provide a black and white DAG in the revised version of the manuscript.

Results P6L125: In your revised analysis, the attenuation of the OLS estimates on adjusting for control variables should no longer be described as “slight” – from 0.037 SD to 0.014 and from 0.023 SD to 0.009 is quite a big reduction, at least on a relative scale.

Thank you. We deleted the word “slightly”.

Results P6L131: The estimate for the unadjusted OLS model in mothers is 12.6% in Table 1 and 13.6% here – please check.

Thank you. The table is correct. We corrected the number in the text.

Results P6L130-132: These are huge estimates – The paternal age result suggests that the educational attainment of the child of a 40-year-old father would be four standard deviations higher than for the child of a 20-year-old father! I think it would be worth commenting on this (implausible?) magnitude in the Discussion and considering why they are so much higher than the OLS estimates, if you have any suggestions.

We do not think that these estimates are reliable because they do not control for the important confounding influences in our DAG. We discuss this now in the Discussion section on pages 7-8: “Specification analyses reveal that the positive association arises from a null effect once the children’s PGI and correlations between different PGIs are taken into account. In an IV model without further controls, we observe a large positive effect of parental age, which we consider implausible and unreliable. This effect is likely due to unobserved confounding. Controlling for children’s PGI for AFB reduces the IV estimate. Finally, we observe a strong change in the IV estimate when we include the PGI for parental educational attainment. This is unsurprising as the genetic association between the PGI for education and the PGI for AFB is well established and the education PGI is known to be one of the strongest PGIs in behavioral science.”

Results P6L132-133: Please can you report correlations among the control variables and between control variables and AFB PGI in parents? It would be particularly helpful to know how the AFB PGI is associated with the other PGIs. This could be a correlation matrix in the supplement.

We now provide a correlation matrix as Table S3 in the *Online Supplement*. Please see also our reply to your point above.

Results P7 L142: Please specify that the influential inherited genes here are specifically the genes included in the educational attainment PGI.

We specify this now on pages 7-8: “Finally, we observe a strong change in the IV estimate when we include the PGI for parental educational attainment. This is unsurprising as the genetic association between the PGI for education and the PGI for AFB is well established and the education PGI is known to be one of the strongest PGIs in behavioral science.”

Table 1: I think there’s a typo in the LCL for the maximally adjusted OLS estimate in mothers – a missing zero? There’s also the inconsistency between text and table mentioned above for the unadjusted OLS model in mothers.

The inconsistency has been corrected. The mentioned mistake in the CI has been corrected.

Table 1: It’s helpful to have the models numbered like this, but please can you make the numbers match those in Figure 1 (even though that would leave gaps in the numbers in Table 1)? This would make it much easier to see how they correspond.

We matched the numbers. Table 1 reports Models 1, 2, 12, and 13 from Figure 2.

Table 1: I think it would be better to refer to it as an F statistic (or just F), and the notes should say that it’s an F stat for the effect of the instrument on the exposure. In fact, in the adjusted models, this should be the _partial_ F statistic (for the instrument). If this is what you report, please state so.

We made these changes.

Table 1: The BIC here don't match those for either component model in the supplement, so am I right in thinking they are for the combined IV model? The notes should clarify this. Also, I suspect BIC are only comparable within a model type (i.e. you can't compare an IV model with an OLS one) but I'm not sure of this – do you have any evidence one way or the other? If it is the case, it should be noted.

We have added that the BIC of the IV models refers to the 2SLS models to the notes of Table 1. We found no evidence either way that BIC between IV models and OLS models are or are not comparable. We, however, would also rather think that they are not comparable.

Table 1. Thanks for stating the units of the outcome, but the units of the estimates also require units of the exposure, so it would be better to say that the units of the estimates are SD of educational outcome per year of AFB. Units of age might seem obvious, but the reader might wonder if they too are SD.

The exposure is not measured in SD; it is indeed one year of AFB. We added this information

Table 1: You could save space and make the table easier to read by extending “Maternal (/Paternal) age at first birth” across the columns. You could also add the sample size for each parent to this text to save using a whole row for a single value below.

We made these changes. Thank you.

Figure 2: This graph is really helpful in visualising the results, but my immediate response was to sketch the OLS estimates on for comparison – please can you add them?

We added the OLS estimates to Figure 2 in the manuscript.

Figure 2: As mentioned previously, consistent model numbers between Figure 2 and Table 1 would make interpretation easier. It might also help if you used different symbols to distinguish OLS and IV models.

We made these numbers consistent.

Figure 2: Having different scales on the Y axis for mothers and fathers gives a false impression that the estimates are very similar in each parent. Please can you make the scales match?

We matched the scales in Figure 2.

Figure 2: In the Y axis titles, please give the units of the estimates (SD of educational attainment per year of AFB).

We added this information but in the title of the figure, which seemed to us more simple and less repetitive.

Tables S1 & S2: Several of my comments on Table 1 also apply to the supplementary tables.

We also correspondingly updated the tables in the *Online Supplement*.

Tables S1 & S2: There's nothing in the "F-test" rows. If the values in Table 1 are partial F stats for the instrument in predicting the exposure, then I think you could omit these rows from the supplement (or repeat the values in Table S1 only)

We deleted the F-test rows from the supplement.

Tables S1 & S2: Please state what the units of the estimates are. In particular, how are the PGIs scaled?

We added the information to the notes of the tables.

Discussion P7L158-160: Elsewhere (P7L146-148, P2L32-34), you stop short of claiming a negative effect of AFB on offspring educational attainment and emphasize the lack of a positive effect in IV models accounting for direct genetic effects. I think this is wise given the possible complications around adjusting for the education PGI. But if we consider the IV models accounting for direct genetic effects to be null, then this suggests only that non-transmitted parental genetics have a null effect on the outcome (not negative). You didn't add the non-transmitted effect to the transmitted effect (cancelling it out), you removed the transmitted effect from the combined effects. In other words, I don't think you can say that the non-transmitted genetics have an "opposite" effect to the transmitted genetics because the opposite to a positive effect would be a negative effect.

We agree and deleted these sentences from the revised version of the manuscript.

Discussion P8L175-177: A small point, but an effect of parental wealth could be non-causal in terms of AFB as suggested here (lifelong wealth and AFB are both caused by SEP) but it could also be mediating a causal effect (individuals get richer as they get older). If individuals would get richer as they get older and this would be causally affecting child education, we would call this a causal effect of parental age. Because parental age causally comes prior to parental resources and a change in parental ages would lead to a change in parental resources. This is indeed what Powell et al. (2006) argued. However, the findings of our study do not provide evidence for such a dynamic (keeping always in mind that our estimates are imprecise and we find no evidence for an age effect but we cannot rule out that

there is an age effect with the large confidence intervals of our estimates). We re-phrased the paragraph to emphasize these points: “Our results do not support sociological expectations of positive effects of higher maternal and paternal age on children’s education. The positive associations between parental ages and children’s academic performance have been attributed by social scientists to higher socioeconomic status and the greater resources that parents accumulate as they age (Powell et al. 2006). In addition, it has been argued that higher parental age leads to more stable relationships and better parenting (Kalmijn and Kraaykamp 2005), which is beneficial for children’s academic performance. Our results do not provide evidence to support these expectations, although uncertainty remains due to the large confidence intervals.” (Page 8)

Discussion P8L182: I find it hard to extract the intended meaning from “Our findings are rather but also not robustly...”, can you re-phrase?

We changed the wording of this paragraph to: “Our findings also do not support biological theories that predict even negative effects of paternal and maternal age. Biologists have observed potentially deleterious mental health consequences of increasing parental ages, particularly related to paternal accumulation and inheritance of de novo mutations across the life course and increased psychiatric problems in children. In addition, age-related methylation changes (Salameh et al. 2020), which may modify parenting behavior, may significantly influence children's school performance, including episodes of psychosis (Bassett et al. 1996; MacCabe et al. 2009; McGrath et al. 1999). While our results do not directly address these potential mechanisms influencing children’s test scores-and mathematical models have shown that de novo mutations are unlikely to have large effects in general (Gratten et al. 2016), they do not provide evidence for robust negative effects of parental age. The possibility remains that

positive social consequences of parental age effects and negative biological consequences cancel each other out.”

Discussion P10L215: should be “contradicts the idea that” (not “the”) – but see my objections to this statement.

Thank you. We corrected this mistake.

Methods P11L248: There should be no apostrophe after “parents”.

This has been corrected. Thank you.

Methods P11L258-259: “our polygenic indices (PGIs) are based on all these SNPs” – presumably you mean that all these SNPs were candidates for the PGIs, and then those associated with each trait in the discovery sample were used to make the PGI? The current wording could be taken to mean that all 6.9 million SNPs actually contributed numerically to the PGIs – can you clarify this please?

We thank the reviewer for this comment and aimed to clarify that the polygenic index construction is based on these SNPs, not the PGIs themselves, page 11: “Our polygenic index (PGI) constructions are based on all these SNPs.” However, the polygenic index construction is based not only on the main hits of the respective GWAS. Due to polygenicity, thousands of variants contribute to the prediction, also those not identified as main hits in the discovery. All this information is used in a polygenic index. Fertility is furthermore known to be particularly polygenic.

Also, was Mills et al 2021 the discovery study used for the control PGIs as well as for the ADB PGI? I think the text hints at this but it’s not stated explicitly.

Thanks, we needed to clarify this as the Mills et al study has only been used for AFB PGI construction, page 11: “Each PGI construction is based on summary statistics provided by the trait-specific GWAS.”

Methods P12 L262-274: The OLS analysis should be mentioned somewhere in this section.

Thank you. We added to this section the following sentence: “For comparison reasons, we also report OLS regression models that estimate the associations between parental age and child test scores, without controlling for the endogeneity of age at first birth.” (Page 12)

References P14L331: The Mills 2021 citation has a stray “262” in the title and I think the page numbers are probably from an online advance version rather than the final version (which is pp1717-1730).

Thank you. We corrected the reference.

Reviewer 2

1. The increase in the sample size is not discussed in the rebuttal. Both versions of the manuscript state that "Our sample is restricted to first-born children who were born between 2001 and 2008 in Norway", yet the sample size has gone up by over 50%. In re-reading the manuscript it would have been beneficial to understand where these new cases came from and how they could be different from those originally used, especially with the estimates in the full model now significantly negative.

We apologize if we missed to explain the increase in sample size in the previous round of RandR. It was simply due to the availability of new data from the MoBa sample, as more genotypes became available. There was no change in the sample selection criteria that we focused on first born children only.

2. The estimation strategy was changed from using only non-transmitted alleles in making the parental AFB PGI to using the full PGI and controlling for the child's PGI for AFB. In the rebuttal the Authors comment that this change allowed them to disentangle the effects of genetic inheritance from other processes. This discussion has not made it into the manuscript though. I think that the rationale for this modelling choice should be discussed in the manuscript, with a robustness check using the non-transmitted alleles strategy presented in the appendix.

We thank the reviewer for pointing out that this argument has not made it into the manuscript. We now added it on p 12: "We use the parental PGI for AFB as the instrument, while conditioning on the child's PGI for AFB." Unfortunately, for the new and larger dataset, non-transmitted scores are not readily available to us. Another complication with untransmitted scores is that with the software approach we used (PseudoCons), the scores represent both parents and thus the causal structure becomes unclear. We explored an alternative approach of imputing grandparental PGIs. This would let us estimate the IV models using mother's and father's ages in a clear fashion, but would also effectively cut our analytical sample by more than 80%, rendering comparison of results across approaches tenuous. We therefore think that comparing approaches may be beyond the scope of this study, but if the reviewer insists, we can of course attempt to produce such scores (or at least imputed parental scores, which is perhaps a more viable alternative) and provide a sensitivity analysis close to what is requested.

3. I found myself wanting for commentary on what distinguishes the PGI for education from the other PGI controls. My guess was that it captured cognitive performance, but then I reached for the AFB GWAS paper (Mills et al. 2021). They report that cognitive performance was not a mediating trait in the genetic correlation of AFB with education

and further note that "the genetic correlation between years of education and AFB is largely a product of a strong bidirectional relationship between these traits, rather than being both downstream of a common identified cause". In light of the full model tipping negative after the inclusion of this variable, some discussion of why it is a particularly potent control for pleiotropy is warranted.

We apologize for the confusion and agree with the reviewer that clarification is required. On p 4 we state that: "This phenomenon, known as pleiotropy, can be due to shared biological influences, social mediation, or genetic effects on third common causes (Wedow et al. 2018), and can be more pronounced the further a predicted outcome is from actual biological processes-which makes AFB particularly susceptible to pleiotropy." The causal processes we control for include therefore education mediating genetic effects on fertility, for example. The fact that the control is so effective is owed to the large genetic correlation between education and AFB ($r_G > 0.6$) combined with the fact that the education score is comparably powerful in its prediction. We added the following to the manuscript on pages 7-8: "Finally, we observe a strong change in the IV estimate when we include the PGI for parental educational attainment. This is unsurprising as the genetic association between the PGI for education and the PGI for AFB is well established and the education PGI is known to be one of the strongest PGIs in behavioral science."

4. It would be helpful to have a legend for the DAG in Figure 1 that defines colors for nodes and edges.

The colors have no meaning. We have created a new version of the DAG that uses no colors anymore.

5. I liked how the note to Figure 1 in the original manuscript grouped model variables by processes that they are meant to control for. I think that this is somewhat lost in the revised manuscript. Perhaps color-coding the bars in Figure 2 into categories of population stratification, genetic inheritance, assortative mating, and pleiotropy would bring back that information.

We tried ways to address this but unfortunately, we did not have a good solution. We add here a color version of Figure 2. We can include it in the manuscript if you prefer but in our view it is no improvement over the black and white figure. The color codes are: Red, no control, Yellow pop strat, green assortative mating, blue childrens traits, black, pleiotropy.

Figure 2: Color Version

6. "one year later mother's AFB" sounds awkward. Maybe "one year delay in mother's AFB"? Also "with higher performance by 2.3%" could be revised to "with performance rising by 2.3%".

Thank you for these suggestions. We adopted them.

7. In their rebuttal the Authors write that they updated the title and abstract to indicate that the analysis is restricted to firstborn children. These revisions are missing from the June 7 version of the manuscript shared with reviewers. Similarly, the reference to (Nivard et al. 2024) invoked in response to comment 23 does not appear in the manuscript.

The title of the manuscript is “No Evidence of Positive Causal Effects of Maternal and Paternal Age at First Birth on Children’s Test Scores at Age 10”, therefore it includes “first birth”. The abstract includes the sentence: “We use an instrumental variable approach (Mendelian Randomization) using maternal and paternal polygenic indices (PGIs) for age at first birth, while conditioning on the child’s PGI for age at first birth, to identify the causal effects of maternal and paternal age at first birth on children’s test scores using data from the Norwegian Mother, Father and Child Cohort Study (MoBa).” making clear that our study only refers to first born children. We deleted the reference to Nivard et al. (2024) from the manuscript because the research design in Nivard et al. (2024) is different from the one in our paper, we apologize for having missed to delete the reference from the response letter.

Detailed comments from Reviewer #1:

L80: Did you mean “risk of externalising behaviour” (not “or”)?

Our reply: Risk of. Corrected.

L102: This threshold would normally be written for publication with the exponent superscripted.

Our reply: We have changed this.

L125: Because you define “controls” in the table as excluding parental PGI for education, I think you need to write “adding the control variables and parental PGI for education” or at least “adding all the control variables” to describe model 13.

Our reply: We have changed this.

Figure 2: The last bit of the figure title doesn’t quite sound right. Maybe “...different model specifications with covariates added stepwise”.

Our reply: We have changed this.

L144: Similarly to the comment above, I think “...performing analyses with covariates added stepwise” would be better.

Our reply: We have changed this.

Figure 2: Are the CI for the OLS estimates just so small that they don’t show up beyond the thickness of the point? If so, I think this should be mentioned in the table footnote just in case a reader thinks they have been omitted.

Our reply: The CIs are shown. We added the information to the figure.

Table S2: The title is missing the word “age” in “PGIs for at first birth”

Our reply: We have added “age”.

L245: Can you specify which analysis this power calculation is referring to - the OLS or the IV?

Our reply: The IV analysis. We have added this information.